# SCALER: FAST AND EFFECTIVE GRAPH ANOMALY DETECTION VIA DUAL-LEVEL SYNERGISTIC CONTRASTIVE LEARNING

## ABSTRACT

Unsupervised graph anomaly detection (UGAD) is crucial for identifying anomalous behavior in graph-structured data. However, recent deep learning-based UGAD methods, while effective, suffer from long inference times due to neighborhood aggregation, which limits their applicability in real-world scenarios. Moreover, current contrastive-based approaches are constrained by the limitations of node–subgraph contrast and the limited use of edge-level contrastive signals. To address these issues, we propose SCALER, a self-supervised MLP-GNN learning framework that trains a structure-aware multilayer perceptron (MLP) for UGAD without requiring costly graph neural network (GNN) aggregation during inference. SCALER introduces a dual-level contrastive learning network that combines node-level and edge-level contrast to effectively guide MLP training. The edge-level contrastive strategy leverages rich relational information embedded in edges to enhance node representations and improve anomaly detection. Furthermore, a neighborhood entropy-guided anomaly score correction module is incorporated to further improve robustness against anomalous nodes with low neighborhood entropy. Extensive experiments on eight real-world benchmark datasets, including a large-scale OGB dataset, against thirteen state-of-the-art baselines demonstrate that SCALER significantly improves detection performance across three metrics, particularly achieving an average gain of 19.6% in AUPRC, while reducing inference time to the order of seconds. These results validate the effectiveness and efficiency of SCALER.

## 1 INTRODUCTION

Graph anomaly detection (GAD) has attracted significant attention due to its wide range of applications in security-sensitive domains (Qiao et al., 2025; Pazho et al., 2024). Within this area, unsupervised graph anomaly detection (UGAD) plays a central role, as it directly addresses the challenges of label scarcity and the high cost of manual annotation. The objective of GAD is to identify anomalous nodes that deviate significantly from the majority of other nodes in a graph. These anomalies are generally categorized into two main types (Liu et al., 2022a;b): **structural anomalies**, where nodes exhibit abnormal connectivity patterns compared to regular nodes, and **contextual anomalies**, where nodes follow normal structural patterns but have attributes that differ substantially from those of their neighbors.

The advent of graph neural networks (GNNs) in deep learning has substantially improved anomaly detection performance compared with traditional shallow methods such as AMEN (Perozzi & Akoglu, 2016), Radar (Li et al., 2017), and ANOMALOUS (Peng et al., 2018). By capturing high-dimensional features and complex graph structures, deep learning-based methods achieve stronger detection performance. However, their deployment is hindered by high inference latency. Fig. 1 illustrates this issue by comparing inference times on the Citation and DBLP datasets between our method and several deep learning methods that use a sampling-based, multi-round anomaly estimation strategy. As shown in Fig. 2, this inefficiency primarily stems from neighborhood aggregation operations during the inference phase and the complexity of anomaly evaluation strategies. Additional overhead comes from multiple detection rounds needed to reduce sampling randomness. These factors result in low efficiency and limited flexibility in practice, failing to meet the require-

ments of anomaly detection in real-world scenarios. Recent studies (Pan et al., 2025; Li et al., 2025a; Zhou et al., 2025; Chen et al., 2024) focus on improving detection performance while maintaining high inference efficiency. This motivates the following question: **Can we design an anomaly detection framework that achieves both high performance and high inference efficiency?**

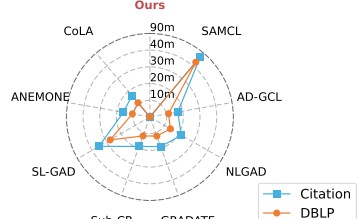

Figure 1: Inference time of various methods on Citation and DBLP.

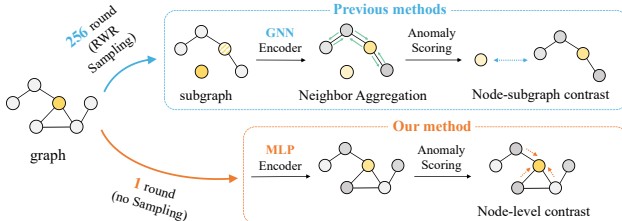

Figure 2: Comparison of the inference phase: previous UGAD methods versus **SCALER**.

Following this question, we further analyze existing deep learning methods and identify two shortcomings in the contrastive learning strategies used in UGAD: **(1) Limitations of sampling-based node–subgraph contrastive learning.** The pioneering contrastive-based UGAD method, CoLA (Liu et al., 2022b), constructs node–subgraph pairs via sampling to model the relationship between a node and its local structure. For UGAD, more representative local structure embeddings are essential for reliable node–neighborhood relationship modeling. Intuitively, the most informative neighborhood of a node consists of its first-order neighbors. Since most graphs exhibit a long-tail degree distribution (i.e., most nodes have low degrees), sampling-based subgraphs often discard important first-order information. Moreover, a randomly chosen negative subgraph for the target node may be inappropriate and even represent a positive instance, making contrast unreliable. To mitigate the randomness introduced by sampling, existing methods often require up to 256 detection rounds, significantly increasing inference time. **(2) The limited exploitation of edge-level contrast.** Many subsequent methods (Jin et al., 2021; Zhang et al., 2022a; Duan et al., 2023a;b; Ma et al., 2025b; Hu et al., 2025) further refine CoLA by incorporating more intricate contrastive strategies. However, these methods mainly focus on the node, subgraph, or graph level, while edges are only considered as auxiliary information. Since edges encode crucial relational information between nodes, the absence of edge-level contrastive information constrains the model's ability to capture complex structural patterns and fully utilize the graph information, ultimately leading to suboptimal performance in GAD.

To address these challenges, we propose **SCALER** (Dual-Level **S**ynergistic **C**ontrastive **A**nomaly **L**earning with **E**nhanced **R**epresentation), a self-supervised UGAD framework via MLP–GNN learning that trains a structure-aware MLP through a dual-level contrastive learning network. Our method incorporates structural knowledge from a GNN into an MLP, producing an MLP encoder that retains structural awareness. This design eliminates the expensive neighbor aggregation during inference, greatly accelerating detection. For node-level contrast, we redefine the local structural positive and negative samples of the target node as all first-order neighbors (positive neighborhood) and all non-first-order neighbors (negative neighborhood) for neighborhood contrastive learning. This design fully leverages first-order neighbor information and ensures more appropriate negative samples than CoLA. It also avoids the need for multiple detection rounds, significantly improving inference efficiency. Overall, this is a comprehensive optimization of CoLA's node–subgraph contrastive framework. At the edge level, we introduce a new contrastive strategy using positive and negative edge pools to exploit edge information. This strategy constructs three sub-positive edge pools and a global negative edge pool for contrastive learning, fully exploiting edge-level contrastive information to enhance anomaly detection. In summary, our key contributions are as follows:

- We propose a self-supervised MLP–GNN learning framework for UGAD, named **SCALER**. It aims to obtain a structure-aware MLP for anomaly estimation, significantly enhancing inference efficiency.

- To guide MLP training, we design a dual-level synergistic contrastive learning network. Specifically, we propose a node-level neighborhood contrastive strategy, which addresses the limitations of traditional node–subgraph contrast. We then propose a novel edge-level

contrastive strategy based on positive and negative edge pools, tackling the largely un-explored edge-level contrastive learning in current UGAD approaches. In addition, we develop a neighborhood entropy-guided anomaly score correction module to enhance robustness against anomalous nodes with low neighborhood entropy.

- Extensive experiments on eight benchmark datasets show that SCALER achieves state-of-the-art performance across multiple evaluation metrics (AUROC, AUPRC, and Recall@K) compared to 13 baseline methods, while also delivering much faster inference.

## 2 PROBLEM FORMULATION

We now provide a detailed description of UGAD. The main notations are summarized in Appendix F. An attributed graph is represented as $\mathcal{G} = (\mathcal{V}, \mathcal{E}, \mathbf{X})$, where $\mathcal{V} = \{v_i\}_{i=1}^N$ denotes the set of $N$ nodes, $\mathcal{E}$ denotes the edge set, and $\mathbf{X} \in \mathbb{R}^{N \times f}$ is the node feature matrix with $f$-dimensional attributes. The $i$-th row of $\mathbf{X}$ (written as $\mathbf{X}_i$) corresponds to the feature vector of node $v_i$. The structural information is encoded by a binary adjacency matrix $\mathbf{A} \in \mathbb{R}^{N \times N}$, where $\mathbf{A}_{i,j} = 1$ if $(v_i, v_j) \in \mathcal{E}$, and $\mathbf{A}_{i,j} = 0$ otherwise. For simplicity, the graph can be represented as $\mathcal{G} = (\mathbf{A}, \mathbf{X})$. In UGAD, ground-truth anomaly labels are unavailable during training. Given the attributed graph $\mathcal{G}$, the objective is to learn a function $\text{score}(\cdot) : (\mathbf{A}, \mathbf{X}) \to \mathbf{S} \in \mathbb{R}^{N \times 1}$ that assigns an anomaly score to each node (with higher scores indicating a higher likelihood of being anomalous).

## 3 METHODOLOGY

The overall pipeline of our method, **SCALER**, is illustrated in Fig. 3. During the training phase, we introduce a self-supervised MLP-GNN learning framework for training a structure-aware MLP, which is then used for subsequent anomaly estimation. The framework consists of two components: (1) a graph encoder with MLPs and GNNs, and (2) a dual-level synergistic contrastive learning network. During the inference phase, we employ the MLP encoder learned during training for anomaly estimation and use neighborhood entropy to correct the anomaly scores. Since anomaly estimation requires only a single computation per node, the inference process remains highly efficient.

### 3.1 GRAPH ENCODER WITH MLPS AND GNNS

To accelerate inference, we aim to train an MLP encoder for anomaly detection. However, relying solely on attribute information neglects structural dependencies, leading to suboptimal performance. To address this, we introduce a self-supervised MLP-GNN framework that incorporates structural knowledge from GNNs into MLPs, thereby effectively integrating attribute and structural information and enabling MLPs to become structure-aware. We sequentially encode the graph $\mathcal{G}$ using an MLP encoder and a GNN encoder to obtain their respective embedding spaces: the MLP embedding space $\mathcal{S}^{\text{MLP}}$ and the GNN embedding space $\mathcal{S}^{\text{GNN}}$.

**MLP as encoder.** We first leverage the graph's attribute information to obtain a structure-free MLP embedding through an MLP encoder with parameters $\theta$:

$$\mathbf{H}^{\text{MLP}} = \mathbf{MLP}_\theta(\mathbf{X}) \tag{1}$$

During the inference phase, we employ this MLP encoder for anomaly estimation to accelerate inference.

**GNN as encoder.** We then incorporate the MLP embedding and the graph's structural information into a GNN to extract a structure-aware GNN embedding:

$$\mathbf{H}^{\text{GNN}} = \mathbf{GNN}(\mathbf{A}, \mathbf{H}^{\text{MLP}}) \tag{2}$$

In practice, we adopt a one-layer graph convolutional network (GCN) for its computational efficiency. Eq. (2) can then be expressed as:

$$\mathbf{H}^{\text{GNN}(\ell)} = \sigma \left( \widetilde{\mathbf{D}}^{-\frac{1}{2}} \widetilde{\mathbf{A}} \widetilde{\mathbf{D}}^{-\frac{1}{2}} \mathbf{H}^{\text{GNN}(\ell-1)} \mathbf{W}^{(\ell-1)} \right) \tag{3}$$

where $\widetilde{\mathbf{A}} = \mathbf{A} + \mathbf{I}$ is the adjacency matrix with self-loops, $\widetilde{\mathbf{D}}$ is the degree matrix of the graph, $\mathbf{W}^{(\ell-1)}$ is the weight matrix of the $(\ell-1)$-th layer, $\sigma(\cdot)$ is the activation function such as ReLU, and $\mathbf{H}^{\text{GNN}(0)}$ is the MLP embedding $\mathbf{H}^{\text{MLP}}$.

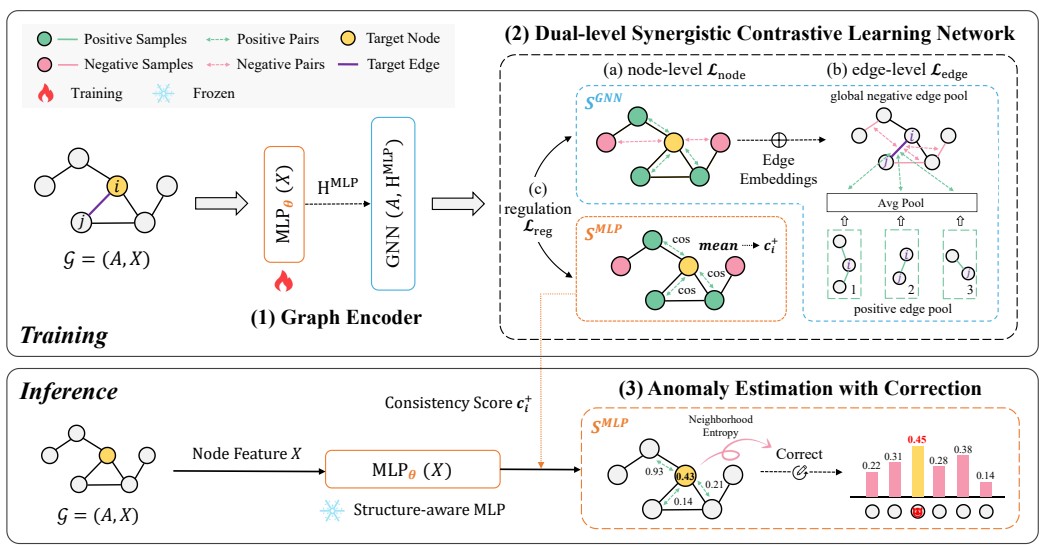

Figure 3: The overview framework of **SCALER**.

## 3.2 DUAL-LEVEL SYNERGISTIC CONTRASTIVE LEARNING NETWORK

### 3.2.1 NODE-LEVEL CONTRASTIVE NETWORK

To address the limitations of CoLA's node-subgraph contrastive strategy—mainly the loss of neighborhood information and the randomness in subgraph sampling—we propose a node-level neighborhood contrastive network. Our method achieves this by redefining positive and negative neighborhood samples for each target node. This approach preserves first-order neighborhood information, avoids mismatched negative samples, and substantially improves inference efficiency.

We discard subgraph sampling for each target node and instead define its positive neighborhood as the set of all first-order neighbors, while its negative neighborhood consists of all non-first-order neighbors. Formally, for a target node $v_i$, its positive and negative neighborhoods are defined as:

$$\mathcal{N}^+(v_i) = \{ v_j \mid v_j \in \mathcal{N}(v_i) \}, \quad \mathcal{N}^-(v_i) = \{ v_j \mid v_j \in \mathcal{B} \setminus \mathcal{N}(v_i) \}. \tag{4}$$

where $\mathcal{N}(v_i)$ denotes the first-order neighbors of $v_i$, and $\mathcal{B}$ represents the set of all nodes in the batch.

In $\mathcal{S}^{\text{GNN}}$, we compute the consistency scores $c_i^+$ and $c_i^-$ for target node $v_i$ by calculating the average cosine similarity between the node and its positive and negative neighborhoods, respectively:

$$c_i^+ = \frac{\sum_{v_j \in \mathcal{N}^+(v_i)} \cos(\mathbf{H}_i^{\text{GNN}}, \mathbf{H}_j^{\text{GNN}})}{|\mathcal{N}^+(v_i)|}, \quad c_i^- = \frac{\sum_{v_j \in \mathcal{N}^-(v_i)} \cos(\mathbf{H}_i^{\text{GNN}}, \mathbf{H}_j^{\text{GNN}})}{|\mathcal{N}^-(v_i)|} \tag{5}$$

where $\mathbf{H}_i^{\text{GNN}}$ represents the embedding of node $v_i$ in $\mathcal{S}^{\text{GNN}}$. To learn a discriminative contrastive learning module, we employ the Jensen–Shannon (JS) divergence (Veličković et al., 2019) as the node-level contrastive network objective function:

$$\mathcal{L}_{\text{node}}^{\text{GNN}} = \frac{1}{2|\mathcal{B}|} \sum_{v_i \in \mathcal{B}} (\log(c_i^+) + \log(1 - c_i^-)) \tag{6}$$

Similarly, we also compute the loss $\mathcal{L}_{\text{node}}^{\text{MLP}}$ in $\mathcal{S}^{\text{MLP}}$ following Eq. (6), which results in our overall node-level contrastive objective with a hyperparameter $\alpha$ controlling the contribution of the loss term from $\mathcal{S}^{\text{MLP}}$:

$$\mathcal{L}_{\text{node}} = \mathcal{L}_{\text{node}}^{\text{GNN}} + \alpha \mathcal{L}_{\text{node}}^{\text{MLP}} \tag{7}$$

### 3.2.2 EDGE-LEVEL CONTRASTIVE NETWORK

To address the lack of edge-level contrast in existing UGAD methods, we propose another contrastive strategy: an edge-level contrastive network based on positive and negative edge pools. This approach constructs pools of positive and negative edge samples to provide richer edge-level contrastive signals, thereby fully exploiting edge information. Although numerous methods exist for edge representation, we adopt a simplified approach by representing an edge embedding as the average of its two endpoint node embeddings. In $\mathcal{S}^{\mathrm{GNN}}$, the embedding of an edge $e_{ij}$ (where $e_{ij}$ and $e_{ji}$ denote the same edge) is defined as:

$$\mathbf{E}_{ij}^{\mathrm{GNN}} = \frac{\mathbf{H}_i^{\mathrm{GNN}} + \mathbf{H}_j^{\mathrm{GNN}}}{2} \tag{8}$$

For the target edge $e_{ij}$, we define its positive edge pool $\mathcal{P}_{ij}^+$ as the set of all edges sharing at least one endpoint with $e_{ij}$, and its negative edge pool $\mathcal{P}_{ij}^-$ as the set of all other edges. Specifically, the positive edge pool $\mathcal{P}_{ij}^+$ is further divided into three sub-pools: $\mathcal{P}_{ij}^{+,1}$, $\mathcal{P}_{ij}^{+,2}$, and $\mathcal{P}_{ij}^{+,3}$. The formal definitions of these positive and negative edge pools for the target edge $e_{ij}$ are:

- $\mathcal{P}_{ij}^{+,1} = \{e_{ij}\}, \quad \mathcal{P}_{ij}^{+,2} = \{e_{ik} \mid k \in \mathcal{N}_i,\ k \neq j\}, \quad \mathcal{P}_{ij}^{+,3} = \{e_{jk} \mid k \in \mathcal{N}_j,\ k \neq i\}$.
- $\mathcal{P}_{ij}^- = \{\, e_{mn} \mid \{m,n\} \cap \{i,j\} = \emptyset \,\}$.

For each positive sub-pool $\mathcal{P}_{ij}^{+,k}$, we compute its embedding via mean pooling:

$$\mathbf{z}_{ij}^k = \frac{1}{|\mathcal{P}_{ij}^{+,k}|} \sum_{e_{mn} \in \mathcal{P}_{ij}^{+,k}} \mathbf{E}_{mn}^{\mathrm{GNN}}, \quad k \in \{1,2,3\} \tag{9}$$

Motivated by the InfoNCE loss, the edge-level contrastive loss associated with the target edge $e_{ij}$ is formulated as:

$$\ell(e_{ij}) = -\log \left( \frac{\sum_{k \in \{1,2,3\}} \exp\left(\mathrm{sim}(\mathbf{E}_{ij}^{\mathrm{GNN}}, \mathbf{z}_{ij}^k)/\tau\right)}{\sum_{k \in \{1,2,3\}} \exp\left(\mathrm{sim}(\mathbf{E}_{ij}^{\mathrm{GNN}}, \mathbf{z}_{ij}^k)/\tau\right) + \sum_{e_{mn} \in \mathcal{P}_{ij}^-} \exp\left(\mathrm{sim}(\mathbf{E}_{ij}^{\mathrm{GNN}}, \mathbf{E}_{mn}^{\mathrm{GNN}})/\tau\right)} \right) \tag{10}$$

where $\mathrm{sim}(\cdot,\cdot)$ denotes cosine similarity and $\tau$ is a temperature parameter.

Due to the large number of edges, the negative edge pool contains an excessive number of samples, which results in high computational overhead and potential GPU memory overflow. To address this issue, we randomly sample a fixed number of negative edges from the edge set $\mathcal{E}$ for each target edge to construct a global negative edge pool $\mathcal{P}_{global}^-$. This global pool replaces the negative edge pool $\mathcal{P}_{ij}^-$ in Eq. (10). However, while the size of the negative pool remains constant across edges, the size of the positive pool varies with node degree, resulting in imbalance. To mitigate this effect and prevent large numerical discrepancies between the edge-level loss and the node-level loss, we scale the loss by the total number of positive samples: $\bar{\ell}(e_{ij}) = \frac{\ell(e_{ij})}{|\mathcal{N}_i| + |\mathcal{N}_j| - 1}$. Finally, the overall loss of the edge-level contrastive network is defined as:

$$\mathcal{L}_{\mathrm{edge}} = \frac{1}{|\mathcal{E}|} \sum_{e_{ij} \in \mathcal{E}} \bar{\ell}(e_{ij}) \tag{11}$$

### 3.2.3 JOINT TRAINING

During training, two contrastive strategies are employed to effectively capture structural information and guide MLP learning. However, these strategies may introduce potential distribution shifts between different representations. To mitigate this issue, we incorporate a reconstruction term as an additional regularization component, which helps reduce distribution shifts and provides more signals for MLP training.

$$\mathcal{L}_{\mathrm{reg}} = \frac{1}{|\mathcal{B}|} \sum_{v_i \in \mathcal{B}} \|\mathbf{X}_i - \mathcal{D}(\mathbf{H}_i^{\mathrm{GNN}})\|^2 \tag{12}$$

where $\mathcal{D}(\cdot)$ can be either identity or learnable. In practice, we adopt a non-linear MLP. Hence, the overall objective function is:

$$\mathcal{L} = \mathcal{L}_{\text{node}} + \beta\mathcal{L}_{\text{edge}} + \gamma\mathcal{L}_{\text{reg}} \tag{13}$$

where $\beta$ and $\gamma$ are trade-off hyperparameters controlling the relative contribution of the loss components.

### 3.3 ANOMALY ESTIMATION WITH NEIGHBORHOOD ENTROPY-GUIDED CORRECTION

#### 3.3.1 ANOMALY ESTIMATION

During the inference phase, we employ the **structure-aware MLP encoder** learned during the training phase to perform anomaly estimation. For each node $v_i$, we compute the positive neighborhood **consistency score** $c_i^+$ using Eq. (5). For anomalous nodes, the consistency scores of their positive pairs tend to be relatively low, as the model fails to accurately capture their corresponding patterns. Based on this observation, the anomaly score for each node $v_i$ is defined as:

$$s_i = 1 - c_i^+ \tag{14}$$

#### 3.3.2 SCORE CORRECTION

Some anomalous nodes exhibit not only abnormal similarity but also a highly skewed distribution of neighbor similarities. Computing the anomaly score using Eq. (14) fails to capture this abnormality, causing such nodes to be ranked between normal and anomalous ones. To improve the robustness of the anomaly score, we introduce a correction function. The corrected anomaly score for node $v_i$ is defined as:

$$\text{score}(i) = s_i \cdot (1 + \psi(t_i)) \tag{15}$$

where $\psi(t_i)$ is the correction function and $t_i$ measures the uniformity of the consistency scores distribution within the neighborhood. The value of $\psi(t_i)$ is constrained to the range $0 \leq \psi(t_i) \leq 0.2$, ensuring that the correction refines the anomaly scores without significantly altering their original values.

In practice, we employ normalized neighborhood entropy to guide the correction of anomaly scores. For each node $v_i$, we compute a probability distribution over its positive neighborhood and the corresponding neighborhood entropy jointly as:

$$q_{ij} = \frac{\exp(\cos(\mathbf{H}_i^{\text{MLP}}, \mathbf{H}_j^{\text{MLP}}))}{\sum_{k\in\mathcal{N}^+(v_i)}\exp(\cos(\mathbf{H}_i^{\text{MLP}}, \mathbf{H}_k^{\text{MLP}}))}, \quad t_i = -\sum_{k\in\mathcal{N}^+(v_i)} q_{ik}\log(q_{ik}) \tag{16}$$

The final corrected anomaly score is computed as:

$$\text{score}(i) = s_i \cdot \left(1 + \phi(1 - \tilde{t}_i)\right), \quad \tilde{t}_i = \frac{t_i}{\log(|\mathcal{N}^+(v_i)|)} \tag{17}$$

where $\phi(\cdot)$ is a linear mapping function, controlling the correction strength.

## 4 EXPERIMENTS

### 4.1 EXPERIMENTAL SETTINGS

**Datasets.** We evaluate **SCALER** on six datasets with synthetic anomalies: Cora, Citeseer, ACM, Citation, DBLP, and the large-scale OGB dataset ogbn-Arxiv, using a widely adopted injection approach (Liu et al., 2022b), and on two datasets with organic anomalies: Reddit and YelpChi. Details of the datasets and anomaly injection methods are provided in Appendix C.

**Baseline Methods and Evaluation Metrics.** We compare **SCALER** with 13 deep UGAD methods, namely DOMINANT (Ding et al., 2019), CoLA (Liu et al., 2022b), ANEMONE (Jin et al., 2021), SL-GAD (Zheng et al., 2023), Sub-CR (Zhang et al., 2022a), GRADATE (Duan et al., 2023a), NL-GAD (Duan et al., 2023b), ADA-GAD (He et al., 2024), GADAM (Chen et al., 2024), DiffGAD (Li et al., 2025a), AD-GCL (Xu et al., 2025), SAMCL (Hu et al., 2025), and HUGE (Pan et al., 2025). All methods are evaluated using AUROC, AUPRC, and Recall@K metrics. All baselines are executed using their original implementations. Appendix D provides additional details on the algorithm, baselines, and hyperparameters.

Table 1: Performance comparison between **SCALER** and baselines with AUROC and AUPRC. The best and second-best results are in **bold** and underlined, respectively. OOM indicates out-of-memory on a 16GB GPU. IMP indicates the average improvement of **SCALER** over the rest.

| Methods | Cora | | Citeseer | | ACM | | Citation | | DBLP | | ogbn-Arxiv | | Reddit | | YelpChi | |
|---|---|---|---|---|---|---|---|---|---|---|---|---|---|---|---|---|
| | AUROC | AUPRC | AUROC | AUPRC | AUROC | AUPRC | AUROC | AUPRC | AUROC | AUPRC | AUROC | AUPRC | AUROC | AUPRC | AUROC | AUPRC |
| DOMINANT | 0.8889 | 0.4013 | 0.8658 | 0.2902 | 0.8902 | 0.1729 | 0.8671 | 0.3693 | 0.8904 | 0.4356 | OOM | OOM | 0.5542 | 0.0370 | OOM | OOM |
| CoLA | 0.8733 | 0.4141 | 0.8818 | 0.3882 | 0.8948 | 0.3085 | 0.8020 | 0.2897 | 0.7947 | 0.2826 | OOM | OOM | 0.5948 | 0.0409 | 0.4124 | 0.0421 |
| ANEMONE | 0.8922 | 0.4472 | 0.9027 | 0.4784 | 0.9011 | 0.3355 | 0.7970 | 0.2537 | 0.8048 | 0.3023 | OOM | OOM | 0.5930 | 0.0417 | 0.4107 | 0.0420 |
| SL-GAD | 0.8991 | 0.5323 | 0.8946 | 0.5064 | 0.8659 | 0.2995 | 0.8181 | 0.3777 | 0.8454 | 0.4172 | OOM | OOM | 0.5662 | 0.0369 | 0.4067 | 0.0432 |
| Sub-CR | 0.8982 | 0.5453 | 0.9242 | 0.5544 | 0.8670 | 0.3058 | 0.7879 | 0.2992 | 0.7888 | 0.2976 | OOM | OOM | 0.5656 | 0.0370 | 0.4293 | 0.0443 |
| GRADATE | 0.8860 | 0.5189 | 0.8744 | 0.2660 | 0.9090 | 0.3161 | 0.7596 | 0.1764 | 0.7743 | 0.2828 | OOM | OOM | **0.5980** | 0.0423 | OOM | OOM |
| NLGAD | 0.9097 | 0.3906 | 0.9365 | 0.4970 | 0.9217 | 0.2766 | 0.8033 | 0.1996 | 0.8199 | 0.2919 | OOM | OOM | 0.5855 | 0.0399 | 0.4443 | 0.0434 |
| ADA-GAD | 0.8320 | 0.3387 | 0.8649 | 0.3443 | OOM | OOM | 0.8697 | 0.3203 | 0.8944 | 0.3543 | OOM | OOM | 0.5689 | 0.0406 | OOM | OOM |
| GADAM | 0.9461 | 0.7577 | 0.9568 | 0.7096 | 0.9551 | 0.4432 | 0.8615 | 0.3778 | 0.8651 | 0.4175 | 0.8121 | 0.2349 | 0.5822 | 0.0485 | 0.5234 | 0.0633 |
| DiffGAD | 0.8912 | 0.3867 | 0.9173 | 0.3237 | OOM | OOM | 0.8969 | 0.3782 | 0.8899 | 0.4356 | OOM | OOM | 0.5573 | 0.0371 | 0.5626 | 0.0659 |
| AD-GCL | 0.9163 | 0.4933 | 0.9441 | 0.5590 | 0.9467 | 0.4026 | 0.7671 | 0.1932 | 0.7841 | 0.2124 | OOM | OOM | 0.5751 | 0.0394 | OOM | OOM |
| SAMCL | 0.9384 | 0.5838 | 0.9501 | 0.5916 | 0.9353 | 0.3519 | 0.8564 | 0.3422 | 0.8561 | 0.3343 | OOM | OOM | 0.5857 | 0.0395 | 0.4969 | 0.0554 |
| HUGE | 0.9537 | 0.7056 | 0.9693 | 0.7780 | 0.9485 | 0.4623 | 0.9046 | 0.4804 | 0.9049 | 0.5345 | OOM | OOM | 0.5906 | 0.0509 | 0.6015 | 0.0718 |
| **SCALER** | **0.9666** | **0.8307** | **0.9804** | **0.7918** | **0.9621** | **0.5035** | **0.9557** | **0.6437** | **0.9326** | **0.6646** | **0.9050** | **0.3108** | 0.5937 | **0.0523** | **0.6211** | **0.0806** |
| **IMP**(%) | 6.47 | 32.95 | 6.64 | 30.82 | 4.98 | 16.94 | 12.56 | 33.16 | 9.32 | 31.09 | 9.29 | 7.59 | 1.55 | 1.14 | 14.47 | 2.82 |

## 4.2 COMPARISON WITH BASELINE METHODS

We evaluate **SCALER** across three standard metrics on eight datasets against 13 baselines. Tab. 1 reports the AUROC and AUPRC results, while additional detailed experimental results, including Recall@K and standard deviations, are provided in Appendix E.1. From the comparison, we make the following observations:

- **SCALER** demonstrates strong detection performance, achieving the best performance on most evaluation metrics across eight datasets. Notably, on the Reddit dataset, the AUROC score is slightly below the best but remains highly competitive, while the AUPRC score achieves the highest performance among all methods. Across datasets, compared with the best-performing baseline, AUROC improves by 0.7%–11.4%, while AUPRC achieves a more substantial improvement of 1.8%–33.9%, highlighting the method's notable advantage in identifying anomalous samples.

- **SCALER** exhibits notable scalability, efficiently handling large-scale OGB datasets where most baseline methods encounter out-of-memory issues. Compared to baselines, our method does not rely on subgraph sampling or other memory-intensive operations on the entire graph.

## 4.3 ABLATION STUDIES

### 4.3.1 CONTRASTIVE STRATEGY ANALYSIS

To evaluate the effectiveness of the two contrastive strategies in **SCALER**, we conducted comprehensive ablation studies, and the results are summarized in Tab. 2. Among them, **w/ node contrast** indicates that only the node-level contrastive strategy is used to guide MLP learning in the MLP-GNN framework without any other components to facilitate subsequent performance comparison with the CoLA framework, whereas **w/o edge contrast** indicates that the edge-level contrastive strategy is removed from the full set of components. The results show that the complete **SCALER** consistently achieves the best performance across all datasets, confirming that each module contributes positively to the GAD task.

For **w/ node contrast**, the results clearly demonstrate that MLP-GNN learning framework guided by node-level contrastive strategy substantially outperforms CoLA in detection performance. This improvement is largely due to the redefinition of positive and negative sample pairs for each target node, representing a significant advancement over the traditional node–subgraph contrastive ap-

Table 2: Ablation studies of contrastive strategies w.r.t. AUROC and AUPRC.

| Methods | Cora | | Citeseer | | ACM | | Citation | | DBLP | | Reddit | | YelpChi | |
|---|---|---|---|---|---|---|---|---|---|---|---|---|---|---|
| | AUROC | AUPRC | AUROC | AUPRC | AUROC | AUPRC | AUROC | AUPRC | AUROC | AUPRC | AUROC | AUPRC | AUROC | AUPRC |
| **SCALER** | **0.9666** | **0.8307** | **0.9804** | **0.7918** | **0.9621** | **0.5035** | **0.9557** | **0.6437** | **0.9326** | **0.6646** | 0.5937 | **0.0523** | **0.6211** | **0.0806** |
| CoLA | 0.8733 | 0.4141 | 0.8818 | 0.3882 | 0.8948 | 0.3085 | 0.8020 | 0.2897 | 0.7947 | 0.2826 | **0.5948** | 0.0409 | 0.4124 | 0.0421 |
| w/ node contrast | 0.9512 | 0.7564 | 0.9615 | 0.7054 | 0.9360 | 0.4355 | 0.9205 | 0.5196 | 0.9216 | 0.5629 | 0.5863 | 0.0512 | 0.6003 | 0.0736 |
| w/o edge contrast | 0.9516 | 0.7585 | 0.9725 | 0.7349 | 0.9378 | 0.4513 | 0.9251 | 0.5249 | 0.9290 | 0.5749 | 0.5897 | 0.0501 | 0.6096 | 0.0755 |

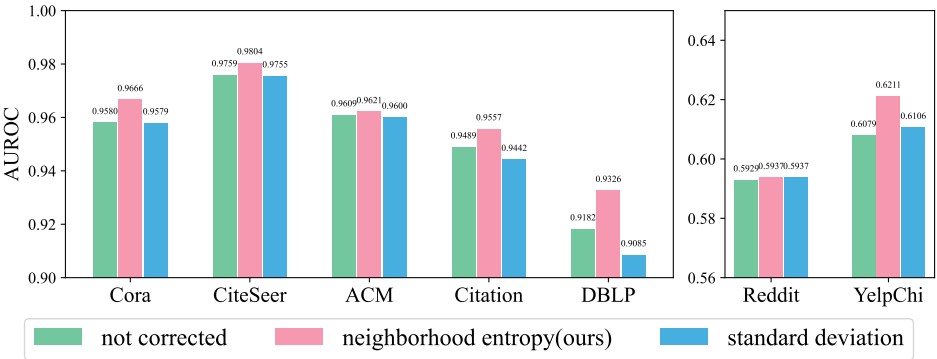

Figure 4: Impact of anomaly score correction. Ablation study comparing different anomaly score correction strategies (none, neighborhood entropy, and standard deviation of neighbor scores) on AUROC. Our entropy-based correction yields higher AUROC, indicating it better separates anomalies (as further discussed in Appendix E.4).

proach employed by CoLA. The model captures more informative local structural embeddings and learns more discriminative node-local relationships. For **w/o edge contrast**, the removal of this component results in a noticeable drop across multiple metrics, especially AUPRC. By introducing edge-level contrastive learning, it fully leverages edge information rather than treating edges merely as auxiliary signals for feature updates. Similar to node-level contrastive learning, this strategy captures the consistent relationships between each target sample and its neighborhood from the edge perspective, providing rich structural information for anomaly estimation. Overall, by employing dual-level synergistic contrastive learning for graph anomaly detection, **SCALER** effectively leverages both node attributes and graph structural information and captures rich relational patterns.

### 4.3.2 ANOMALY SCORE CORRECTION ANALYSIS GUIDED BY DIFFERENT INDICATORS

Some anomalous nodes show a skewed distribution of neighbor similarities. These nodes often attempt to evade detection by establishing connections with normal nodes. Thus, anomaly estimation based solely on first-order neighborhoods may dilute the anomaly signal and fail to effectively capture such nodes. This motivates the use of indicator to quantify the skewness in neighbor consistency scores. Different choices of indicators can lead to varying impacts on detection performance. To investigate their effects, we considered three variations: **not corrected**, where no adjustment is applied; **neighborhood entropy**, which leverages the entropy of neighbor scores; and **standard deviation**, which uses the standard deviation of neighbor scores. The results are illustrated in Fig. 12. Compared to the uncorrected version, using standard deviation fails to provide additional discriminative signals and may even degrade performance. In contrast, the proposed neighborhood entropy-guided anomaly score correction effectively captures irregularities within the neighborhood structure, thereby enhancing detection capability. This improvement is due to neighborhood entropy measuring the disorder among a node's neighbors, making it more sensitive to subtle structural irregularities caused by some low-entropy anomalous nodes. Overall, this entropy-guided correction improves anomaly detection across multiple datasets, with particularly notable gains on Cora, Citation, DBLP, and YelpChi datasets.

Table 3: Efficiency comparison of **SCALER** in terms of inference time (in seconds). IMP $k\times$ indicates that our method is $k$ times faster than the most efficient baseline among the time-intensive inference methods.

| Category | Methods | ACM | Citation | DBLP | Reddit | YelpChi | ogbn-Arxiv |
|---|---|---|---|---|---|---|---|
| | CoLA | 998 | 1,000 | 665 | 647 | 4,800 | OOM |
| | ANEMONE | 903 | 967 | 638 | 657 | 4,840 | OOM |
| | SL-GAD | 2,181 | 2,109 | 1,648 | 1,560 | 10,269 | OOM |
| Time-Intensive | Sub-CR | 1,329 | 1,127 | 726 | 810 | 4,372 | OOM |
| Inference Methods | GRADATE | 1,251 | 1,132 | 729 | 1,556 | OOM | OOM |
| | NLGAD | 2,853 | 1,305 | 852 | 758 | 6,263 | OOM |
| | AD-GCL | 998 | 1,015 | 680 | 1,339 | OOM | OOM |
| | SAMCL | 10,382 | 5,282 | 3,238 | 6,280 | 13,648 | OOM |
| | DOMINANT | 19.990 | 5.653 | 2.037 | 0.789 | OOM | OOM |
| Time-Efficient | ADA-GAD | OOM | 29.920 | 12.566 | 9.334 | OOM | OOM |
| Inference Methods | GADAM | 0.076 | 0.041 | 0.039 | 0.016 | 0.032 | 0.011 |
| | DiffGAD | OOM | 123.251 | 79.677 | 0.510 | 0.752 | OOM |
| | HUGE | 0.062 | 0.042 | 0.026 | 0.037 | 0.066 | OOM |
| Ours | **SCALER** | 0.057 | 0.025 | 0.014 | 0.055 | 0.034 | 0.483 |
| | **IMP(x$10^3$)** | 16× | 39× | 46× | 12× | 129× | - |

## 4.4 EFFICIENCY COMPARISON

Tab. 3 further illustrates the efficiency of **SCALER** in terms of inference time (in seconds). **SCALER** runs significantly faster than its competing methods in time-intensive inference methods, achieving speedups ranging from $12\times$ to $129\times$ across various datasets. Our method and all time-efficient inference methods are able to complete inference within a few minutes, with our method further achieving second-level inference even on the large-scale OGB dataset. However, as shown in Fig. 5 (Appendix E.2), **SCALER** achieves higher performance across two metrics, highlighting both the effectiveness and efficiency of our method. Overall, compared to other deep learning methods, our method not only achieves state-of-the-art performance but also attains high inference efficiency, demonstrating the practicality of the MLP-GNN learning framework and the effectiveness of our proposed dual-level contrastive learning. This represents a substantial improvement over traditional contrastive-based UGAD methods.

## 4.5 MORE EXPERIMENTS

We also conduct additional experiments, including hyperparameter analysis of $\alpha$, $\beta$, $\gamma$ and the number of negative edges (Appendix E.3), visualization of the anomaly learning process and the distribution of anomaly scores with correction (Appendix E.4), generalization analysis of the structure-aware MLP (Appendix E.5), and performance analysis of contextual and structural anomaly detection (Appendix E.6).

## 5 CONCLUSION

In this paper, we analyze the shortcomings of existing unsupervised graph anomaly detection methods. We propose **SCALER**, a MLP–GNN learning framework that trains a structure-aware MLP for anomaly detection, eliminating costly GNN aggregation during the inference phase. To achieve this, **SCALER** introduces a dual-level synergistic contrastive learning network that guides MLP training, while additionally leveraging neighborhood entropy to correct anomaly scores during inference. Extensive experiments across multiple datasets demonstrate that **SCALER** significantly outperforms existing methods, achieving superior detection accuracy while slashing inference times by orders of magnitude.

## 6 REPRODUCIBILITY STATEMENT

The datasets used in our experiments, along with the anomaly injection methods, are detailed in Appendix C. Comprehensive implementation details, including the algorithm implementation and our code, are provided in Appendix D. To further facilitate reproducibility, we also provide the environment configuration and hyperparameter settings in Appendix D.

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

## A   THE USE OF LARGE LANGUAGE MODELS (LLMS)

In this paper, LLMs were used to polish writing for clarity. All ideas, methods, and results presented are our original contributions.

## B   RELATED WORK

### B.1   TRADITIONAL UNSUPERVISED GRAPH ANOMALY DETECTION

The goal of graph anomaly detection (GAD) is to identify rare patterns that deviate from the majority patterns in a graph. Current unsupervised graph anomaly detection (UGAD) methods fall into two main categories: contrastive-based methods and reconstruction-based methods.

Reconstruction-based methods employ graph autoencoders to reconstruct node attributes and adjacency matrices, and use reconstruction errors as anomaly scores. The pioneering work DOMINANT (Ding et al., 2019) introduces GCN-based autoencoders to jointly model structural and attribute information. Recent approaches (Fan et al., 2020; Roy et al., 2024; He et al., 2024) further enhance performance by designing more robust frameworks to alleviate overfitting to anomalies.

For contrastive-based methods, CoLA (Liu et al., 2022b) was the first method to introduce contrastive learning into UGAD, sampling and forming node–subgraph positive and negative sample pairs to model the relationship between nodes and their local structure. Subsequent methods have improved upon the CoLA framework, enhancing performance by designing more complex contrastive strategies and optimizing detection strategies. ANEMONE (Jin et al., 2021) incorporates node–node contrast within the contrastive network. SL-GAD (Zheng et al., 2023) further combines node attribute reconstruction with node–subgraph contrastive learning. Sub-CR (Zhang et al., 2022a) first proposed a multi-view graph augmentation strategy to enhance contrastive learned representations. GRADATE (Duan et al., 2023a) utilizes a multi-scale, multi-view contrastive framework and performs subgraph-level alignment, while SAMCL (Hu et al., 2025) enhances subgraph-level contrastiveness through Earth Mover's Distance. NLGAD (Duan et al., 2023b) improves normality modeling through hybrid selection, and AD-GCL (Xu et al., 2025) boosts tail-node detection with anomaly-guided neighbor completion. However, existing GCL strategies in GAD mainly focus on node-level and subgraph-level contrast and fail to explore edge-level contrast, using edges solely as auxiliary information for feature updates. Moreover, they overlook the intrinsic limitations of CoLA's node–subgraph contrastive strategy.

To enhance node representations, some GAD and self-unsupervised methods (Liu et al., 2024; Li et al., 2025b) focus on modeling edge information for a more comprehensive utilization of graph structure. Our proposed dual-level synergistic contrastive network fills the edge-level contrast gap for GAD tasks. These complementary perspectives make anomalous patterns more distinguishable and improve the detection performance.

### B.2   UNSUPERVISED GRAPH ANOMALY DETECTION WITH EFFICIENT INFERENCE

Most existing methods in GAD follow the contrastive framework of CoLA, which requires GNN aggregation and multiple rounds of subgraph sampling during inference. This leads to slow inference and limits their applicability in real-world settings.

In recent years, several methods have aimed to balance detection performance with inference efficiency. DiffGAD (Li et al., 2025a) integrates discriminative content through diffusion sampling while preserving multi-scale information with low computational cost. TFGAD (Zhou et al., 2025) achieves training-free anomaly detection by separately processing node attributes and local structures via SVD and applying a lightweight scoring function. GADAM (Chen et al., 2024) employs two MLPs for attribute and structural encoding and resolves inconsistencies through adaptive message passing. HUGE (Pan et al., 2025) reduces heterogeneity effects with globally guided ranking

loss and predicts anomaly scores using only an MLP. These works suggest that MLPs provide a direction for UGAD, which can attain both high efficiency and strong detection performance.

### B.3 INFERENCE ACCELERATION ON GNNS

GNN inference acceleration techniques typically include quantization, pruning, and knowledge distillation (KD) (Ma et al., 2025a). Quantization compresses continuous values into compact numerical representations, while pruning removes redundant parameters to reduce computational overhead. KD transfers knowledge from a large teacher model to a smaller student model. Since message passing is the primary computational bottleneck during inference, several recent studies explore transferring the knowledge of trained GNNs into MLPs. For example, GLNN (Zhang et al., 2022b) adopts a standard KD pipeline in which an MLP student mimics a GNN teacher, enabling graph-free inference but inevitably weakening the student's ability to capture structural information. Graph-MLP incorporates structural cues through a neighbor-contrastive loss. NOSMOG (Tian et al., 2023) strengthens structural awareness by injecting positional features and enforcing representational similarity with GNN teachers. GraphECL (Xiao et al., 2024)guides the MLP to learn neighborhood structural distributions distilled from a GNN. SimMLP (Wang et al., 2025) introduces a self-supervised framework that aligns GNN and MLP embeddings to inject structural information into MLPs, enabling GNN-level performance with dramatically faster inference.

Motivated by KD, we depart from the CoLA framework and instead apply a distillation-based paradigm to GAD. Through our MLP–GNN joint training framework, we obtain a structure-aware MLP for anomaly evaluation, completely eliminating the time costs associated with GNN aggregation and multiple sampling.

## C DATASET DETAILS

### C.1 ANOMALY INJECTION DETAILS

We inject two common types of anomalies into the datasets: contextual anomalies and structural anomalies following widely adopted injection approach (Liu et al., 2022b).

**Contextual anomalies.** Contextual anomalies are generated by perturbing node attributes. Specifically, $m$ nodes are randomly selected as contextual anomaly candidates. For each selected node $v_i$, we sample $n$ candidate nodes (typically $n = 50$) and compute the Euclidean distance between $v_i$ and each candidate. The attributes of $v_i$ are then replaced with those of the candidate node that exhibits the maximum distance. For the datasets Cora, Citeseer, ACM, Citation, DBLP, and ogbn-Arxiv, the numbers of contextual anomalies are set to 75, 75, 300, 225, 150, and 3000, respectively.

**Structural anomalies.** Structural anomalies are injected by perturbing the graph topology. First, $q$ nodes are randomly selected and interconnected to form a fully connected subgraph. This process is repeated $p$ times, and the resulting $p \times q$ nodes are labeled as structural anomalies. Across the datasets, $q$ is fixed at 15, while $p$ is set to 5, 5, 20, 15, 10, and 200 for Cora, Citeseer, ACM, Citation, DBLP, and ogbn-Arxiv, respectively.

### C.2 ADDITIONAL DATASET INFORMATION

We conduct experiments on six datasets injected with synthetic anomalies: Cora, Citeseer, ACM, Citation, DBLP, and ogbn-Arxiv, as well as two datasets with organic anomalies: Reddit and YelpChi. For datasets without originally labeled anomalies, we inject anomalies following the same approach as previous works. Detailed statistics are provided in Tab. 4.

- **DBLP (Duan et al., 2023b)** is an author network where an edge exists between two authors if they have coauthored a paper.
- **Cora (Liu et al., 2022b), Citeseer (Liu et al., 2022b), Citation (Duan et al., 2023b), and ACM(Chen et al., 2024)** are publication network datasets composed of scientific publications. Each paper is represented as a node, and citation relationships form the edges of the network.

Table 4: The statistics of datasets.

| Dateset | Nodes | Edges | Attributes | Anomalies | Anomaly Ratio |
|---|---|---|---|---|---|
| Cora | 2,708 | 5,429 | 1,433 | 150 | 5.5% |
| Citeseer | 3,327 | 4,732 | 3,703 | 150 | 4.5% |
| ACM | 16,484 | 71,980 | 8,337 | 600 | 3.6% |
| Citation | 8,935 | 15,098 | 6,775 | 450 | 5.0% |
| DBLP | 5,484 | 8,117 | 6,775 | 300 | 5.5% |
| ogbn-Arxiv | 169,343 | 1,166,243 | 128 | 6000 | 3.5% |
| Reddit | 10,984 | 168,016 | 64 | 366 | 3.3% |
| YelpChi | 23,831 | 98,630 | 32 | 1217 | 5.1% |

- **ogbn-Arxiv (Chen et al., 2024)** is a large-scale citation network where nodes represent papers with 128-dimensional features derived from title and abstract word embeddings, pre-trained using the skip-gram model on the MAG corpus.

- **Reddit (Pan et al., 2025)** is a user–subreddit graph from the Reddit platform. Each user has a binary label indicating whether they were banned, with banned users treated as anomalies. Post texts are converted into feature vectors, and node features for users and subreddits are obtained by aggregating the features of their posts.

- **YelpChi (Pan et al., 2025)** is a user–business review network derived from Yelp. Users and businesses are represented as nodes, with edges indicating that a user reviewed a business. Reviews are transformed into feature vectors, and anomalous users or businesses are identified based on abnormal review patterns or ratings, following prior studies.

# D    IMPLEMENTATION DETAILS

## D.1    ALGORITHM IMPLEMENTATION AND COMPLEXITY ANALYSIS

The general procedures of the **SCALER** framework are shown in Algorithm 1. Our code is available at `https://anonymous.4open.science/r/SCALER-1q_6zxpo/` for verification.

**Time Complexity.**    We discuss the time complexity of each component in **SCALER** respectively. We denote the number of nodes as $N$, the number of edges as $M$, the original feature dimension as $d$, the hidden feature dimension as $d'$, the number of negative edges as $K$, and the number of nodes in the batch $\mathcal{B}$ as $|\mathcal{B}|$. In the training phase, **SCALER** consists of four components: graph encoder, node-level contrast, edge-level contrast, and reconstruction regularization. For graph encoder, the forward pass through MLP layers and GNN layers have a time complexity of $\mathcal{O}\big(Nd'(d + d')\big)$ and $\mathcal{O}\big(d'(Nd' + M)\big)$ respectively. To compute node-level contrastive loss, the complexity is $\mathcal{O}(N|\mathcal{B}|d')$. To compute edge-level contrastive loss, the complexity is $\mathcal{O}(MKd')$. To compute reconstruction regularization loss, the complexity is $\mathcal{O}\big(N((d')^2 + dd' + d)\big)$. To sum up, the total time complexity in training phase is $\mathcal{O}\big(d'(N(d + d' + |\mathcal{B}|) + MK)\big)$. In the inference phase, the time complexity of computing anomaly scores is $\mathcal{O}(Md')$ and that of score correction is $\mathcal{O}(M)$. Thus, the total time complexity in inference phase is $\mathcal{O}\big(d'(N(d + d') + M)\big)$.

**Space Complexity.**    We also consider the space complexity of **SCALER**. We denote the number of nodes in the batch $\mathcal{B}$ as $|\mathcal{B}|$, the number of edges in this batch as $m$ (which depends on $\mathcal{B}$), the number of negative edges as $K$, and the hidden feature dimension as $d'$. The space complexity mainly comes from node-level contrast and edge-level contrast. For node-level contrast, the main memory consumption comes from storing the node feature matrix and the node similarity matrix, so the space complexity is $O(|\mathcal{B}|d' + m + |\mathcal{B}|m)$. For edge-level contrast, the main memory consumption comes from the node feature matrix, edge embedding matrix, and negative sample similarity matrix, so the space complexity is $O(|\mathcal{B}|d' + md' + mK)$. To sum up, the total space complexity is $Max\big(O(|\mathcal{B}|d' + m + |\mathcal{B}|m), O(|\mathcal{B}|d' + md' + mK)\big)$.

---

**Algorithm 1** The proposed **SCALER** Algorithm

---

**Input**: Attributed Graph $\mathcal{G} = (\mathcal{V}, \mathcal{E}, \mathbf{X})$, Training epochs $E$, Batch size $B$, Trade-off parameter $\alpha$, $\beta$ and $\gamma$, Learning rate $lr$, Number of negative edges $K$.
**Output**: Anomaly score of all nodes.

---

1: // *Training phase.*
2: **for** $epoch = 1, ..., E$ **do**
3:     $\mathcal{B} \leftarrow$ Randomly split $V$ into batches of size $B$.
4:     **for** $\mathcal{G}_i = (\mathcal{V}_i, \mathcal{E}_i, \mathbf{X}_i, \mathbf{A}_i) \in \mathcal{B}$ **do**
5:         $\mathbf{H}^{\text{MLP}}, \mathcal{S}^{\text{MLP}} \leftarrow \mathbf{MLP}_\theta(\mathbf{X}_i)$
6:         $\mathbf{H}^{\text{GNN}}, \mathcal{S}^{\text{GNN}} \leftarrow \mathbf{GNN}(\mathbf{H}^{\text{MLP}}, \mathbf{A}_i)$
7:         // *Node-level contrastive learning.*
8:         Calculate the consistency score of each node $c_i^+$ and $c_i^-$ with its positive and negative neighbors via Eq. (5) and node-level contrastive loss $\mathcal{L}_{\text{node}}^{\text{GNN}}$ and $\mathcal{L}_{\text{node}}^{\text{MLP}}$ in the $\mathcal{S}^{\text{GNN}}$ and $\mathcal{S}^{\text{MLP}}$ via Eq. (6), respectively.
9:         $\mathcal{L}_{\text{node}} \leftarrow \mathcal{L}_{\text{node}}^{\text{GNN}} + \alpha \mathcal{L}_{\text{node}}^{\text{MLP}}$
10:        // *Edge-level contrastive learning.*
11:        $\mathbf{E}^{\text{GNN}} \leftarrow$ Compute embedding of edges.
12:        Construct the positive and negative edge pools, and employ simple random sampling to select $K$ edges as global negative samples from the edge pool.
13:        Calculate edge-level contrastive loss $\mathcal{L}_{\text{edge}}$ via Eq. (10)(11).
14:        Calculate regularization loss $\mathcal{L}_{reg}$ via Eq. (12).
15:        $\mathcal{L} \leftarrow \mathcal{L}_{\text{node}} + \beta \mathcal{L}_{\text{edge}} + \gamma \mathcal{L}_{\text{reg}}$
16:        Back-propagate $\mathcal{L}$ to update the parameters of MLP and GNN with learning rate $lr$.
17:     **end for**
18: **end for**
19: // *Inference phase.*
20: $\mathbf{H}^{\text{MLP}} \leftarrow \mathbf{MLP}_\theta(\mathbf{X})$
21: Calculate the anomaly scores $s$ using $\mathbf{H}^{\text{MLP}}$ via Eq. (14).
22: Obtain final anomaly score using neighbor entropy to guide anomaly score correction via Eq. (15).

---

## D.2   BASELINE METHODS

We compare **SCALER** with 13 state-of-the-art deep learning methods for unsupervised graph anomaly detection (UGAD). The details of these baselines are described below.

- **DOMINANT (Ding et al., 2019)** is a deep learning-based graph anomaly detection (GAD) method that employs a graph autoencoder to reconstruct both adjacency and feature matrices. Anomalies are identified by measuring reconstruction errors.

- **CoLA (Liu et al., 2022b)** is a contrastive self-supervised GAD approach that captures anomalous patterns by evaluating the consistency between each node and its local subgraph using a GNN-based encoder.

- **ANEMONE (Jin et al., 2021)** further proposes a node-node contrastive approach based on CoLA, which improves the detection performance.

- **SL-GAD (Zheng et al., 2023)** is a self-supervised GAD framework that integrates multi-view contrastive learning with generative attribute reconstruction to enhance anomaly detection performance.

- **Sub-CR (Zhang et al., 2022a)** is a contrastive learning-based method for GAD. It leverages data augmentation to strengthen multi-view contrastive learning and combines it with attribute reconstruction to identify anomalies.

- **GRADATE (Duan et al., 2023a)** employs multi-scale contrastive learning for GAD, simultaneously comparing node-node, node-subgraph, and subgraph-subgraph relations to detect anomalies across different structural levels.

- **NLGAD (Duan et al., 2023b)** is a multi-scale contrastive learning approach based on normality learning. It refines the model using reliable signals from normal nodes to produce more accurate estimates of node normality.

- **ADA-GAD (He et al., 2024)** is a two-stage framework for GAD. It first reduces graph anomalies through anomaly-denoised augmentation and pretraining, then retrains the decoder for detection on the original graph. The method employs multi-level anomaly-denoised pretraining and node anomaly distribution regularization to capture normal patterns and mitigate overfitting and homophily trap effects.

- **GADAM (Chen et al., 2024)** is a two-stage unsupervised GAD framework that decouples local inconsistency mining from message passing. It enhances detection via adaptive message passing and global consistency discernment. Its hybrid attention mechanism enables node-specific message passing based on anomaly score differences and feature similarities, significantly improving performance.

- **DiffGAD (Li et al., 2025a)** is a diffusion-based latent space learning method to enhance anomaly detection. It integrates discriminative content via diffusion sampling and preserve multi-scale information through a content-preservation mechanism.

- **AD-GCL (Xu et al., 2025)** is a graph contrastive learning-based method for anomaly detection. It addresses structural imbalance by introducing neighbor pruning and anomaly-guided neighbor completion strategies.

- **SAMCL (Hu et al., 2025)** is a subgraph-aligned multi-view contrastive learning framework for GAD. It introduces a subgraph–subgraph level contrastive learning mechanism using Earth Mover's Distance (EMD) to measure similarity between nonaligned subgraphs, addressing limitations of prior methods.

- **HUGE (Pan et al., 2025)** is an unsupervised graph fraud detection method. It uses HALO to quantify the graph's heterophily and addresses heterogeneity through a globally guided heterogeneity ranking loss. The method relies solely on an MLP to predict anomaly scores.

## D.3 EVALUATION METRICS

**AUROC.** AUROC (Area Under the Receiver Operating Characteristic Curve) evaluates the model's ability to distinguish between anomalous and normal instances based on predicted anomaly scores. The ROC curve plots the true positive rate against the false positive rate at various thresholds. In this work, anomalies are treated as the positive class. A higher AUROC value, closer to 1, indicates better anomaly detection performance.

**AUPRC.** AUPRC (Area Under the Precision-Recall Curve) measures detection performance for the positive (anomalous) class by integrating precision and recall across different thresholds. It is particularly informative in highly imbalanced settings, such as anomaly detection, where anomalies are rare. A higher AUPRC indicates stronger ability to detect anomalies while minimizing false positives and false negatives.

**Recall@K.** Recall@K measures the model's ability to detect anomalies within the top K ranked instances, where k equals the number of anomalies in each dataset. It represents the proportion of true anomalies found among these top k instances. A higher Recall@K indicates better performance in prioritizing anomaly detection over normal samples.

## D.4 ENVIRONMENT

Our method is implemented with the following libraries and their respective versions: Python 3.9, CUDA version 11.7, PyTorch 2.0.1, DGL 1.1.2+cu117, numpy 1.26.4, torch-scatter 2.1.2+cu117. All the experiments are conducted on a Linux machine equipped with an Intel(R) Xeon(R) Platinum 8255C CPU @ 2.50GHz, 30 GB of RAM, and a NVIDIA Tesla T4 GPU with 16GB of VRAM.

## D.5 HYPERPARAMETER SETTINGS

The search ranges for the parameters and their optimal configurations are presented in Tab. 5 and Tab. 6, respectively. Each experiment is repeated for 5 runs, where the random seed is set to {0, 1,

Table 5: Range of grid search.

| Parameter | Range |
|---|---|
| epoch | {20, 150, 250, 350, 450} |
| learning rate (lr) | {0.001, 0.0005} |
| hidden | 128 (Fixed) |
| $\alpha$ | {0.1, 0.4, 0.7, 1.0} |
| $\beta$ | {0.1, 0.5, 1.0, 2.0, 4.0} |
| $\gamma$ | {0.0, 0.5, 1.0, 2.0, 4.0} |
| $\tau$ | 1 (Fixed) |
| runs | 5 (Fixed) |
| batch | {8192, 1000} |
| neg_num | [10, 150] (step = 10) |

Table 6: Hyperparameters of **SCALER**.

| Dateset | epoch | lr | $\alpha$ | $\beta$ | $\gamma$ | batch | neg_num |
|---|---|---|---|---|---|---|---|
| Cora | 250 | 0.001 | 0.1 | 1.0 | 0.5 | 8192 | 60 |
| Citeseer | 150 | 0.001 | 0.4 | 2.0 | 4.0 | 8192 | 60 |
| ACM | 450 | 0.0005 | 0.1 | 4.0 | 4.0 | 8192 | 60 |
| Citation | 350 | 0.001 | 0.1 | 0.5 | 0.5 | 8192 | 60 |
| DBLP | 350 | 0.0005 | 0.1 | 1.0 | 1.0 | 8192 | 60 |
| ogbn-Arxiv | 20 | 0.001 | 0.1 | 4.0 | 0.5 | 8192 | 60 |
| Reddit | 350 | 0.0005 | 0.1 | 0.5 | 0.0 | 1000 | 60 |
| YelpChi | 350 | 0.001 | 1.0 | 0.1 | 0.0 | 8192 | 60 |

2, 3, 4}. The hyperparameter neg_num denotes the number of sampled edges for the global negative edge pool.

# E MORE RESULTS

## E.1 COMPREHENSIVE PERFORMANCE RESULTS ACROSS MULTIPLE METRICS

The complete performance results, including the mean and standard deviation of AUROC, AUPRC, and Recall@K, are presented in Tab. 7, Tab. 8, and Tab. 9, respectively. In terms of overall performance, our method consistently outperforms the baselines across three evaluation metrics with particularly notable improvements in AUPRC and Recall@K. Although the AUROC score on the Reddit dataset is slightly below the best-performing method, our method achieves the highest results in AUPRC and Recall@K. The improvement in Recall@K suggests that our method assigns significantly higher anomaly scores to true anomalous nodes compared to normal ones, thereby improving detection sensitivity. In terms of stability, our method exhibits strong robustness. Specifically, for nearly all datasets, the standard deviation of each metric is lower than the median standard deviation observed in the baselines. Although there are a few rare cases where notable fluctuations occur, these fluctuations remain small and do not affect reliability.

## E.2 EFFICIENCY COMPARISON

In Fig. 5, we present the comparison of AUROC and AUPRC scores across different methods on the Citation and YelpChi datasets. In addition, we report the training time of all methods, as shown in Fig. 6. Our method remains highly competitive in training time, outperforming most deep learning methods. Thus, our method strikes a favorable balance among training cost, inference speed, and detection performance.

Table 7: Complete results (mean ± std(%)) with AUROC. The best and second-best results are in **bold** and underlined, respectively. OOM indicates out-of-memory on a 16GB GPU.

| Method | Cora | Citeseer | ACM | Citation | DBLP | ogbn-Arxiv | Reddit | YelpChi |
|---|---|---|---|---|---|---|---|---|
| DOMINANT | 0.8889 ± 0.21 | 0.8658 ± 0.32 | 0.8902 ± 0.00 | 0.8671 ± 0.02 | 0.8904 ± 0.01 | OOM | 0.5542 ± 0.48 | OOM |
| CoLA | 0.8733 ± 1.62 | 0.8818 ± 0.63 | 0.8948 ± 0.13 | 0.8020 ± 1.32 | 0.7947 ± 0.86 | OOM | 0.5948 ± 0.77 | 0.4124 ± 1.04 |
| ANEMONE | 0.8922 ± 1.33 | 0.9027 ± 1.61 | 0.9011 ± 0.39 | 0.7970 ± 4.12 | 0.8048 ± 0.61 | OOM | 0.5930 ± 0.18 | 0.4107 ± 0.83 |
| SL-GAD | 0.8991 ± 0.49 | 0.8946 ± 0.92 | 0.8659 ± 0.77 | 0.8181 ± 0.35 | 0.8454 ± 0.35 | OOM | 0.5662 ± 0.71 | 0.4067 ± 0.68 |
| Sub-CR | 0.8982 ± 1.12 | 0.9242 ± 0.85 | 0.8670 ± 0.82 | 0.7879 ± 1.47 | 0.7888 ± 3.74 | OOM | 0.5656 ± 4.87 | 0.4293 ± 7.86 |
| GRADATE | 0.8860 ± 0.31 | 0.8744 ± 0.30 | 0.9090 ± 0.23 | 0.7596 ± 1.37 | 0.7743 ± 0.86 | OOM | **0.5980 ± 1.60** | OOM |
| NLGAD | 0.9097 ± 0.45 | 0.9365 ± 1.25 | 0.9217 ± 0.22 | 0.8033 ± 1.69 | 0.8199 ± 0.99 | OOM | 0.5855 ± 0.61 | 0.4443 ± 0.53 |
| ADA-GAD | 0.8320 ± 0.14 | 0.8649 ± 0.06 | OOM | 0.8697 ± 0.00 | 0.8944 ± 0.00 | OOM | 0.5689 ± 0.01 | OOM |
| GADAM | 0.9461 ± 0.30 | 0.9568 ± 0.40 | 0.9551 ± 0.06 | 0.8615 ± 0.64 | 0.8651 ± 0.56 | 0.8121 ± 0.57 | 0.5822 ± 0.29 | 0.5234 ± 1.93 |
| DiffGAD | 0.8912 ± 0.01 | 0.9173 ± 0.00 | OOM | 0.8969 ± 0.00 | 0.8899 ± 0.00 | OOM | 0.5573 ± 0.31 | 0.5626 ± 1.13 |
| AD-GCL | 0.9163 ± 1.14 | 0.9441 ± 0.33 | 0.9467 ± 0.18 | 0.7671 ± 0.35 | 0.7841 ± 0.60 | OOM | 0.5751 ± 0.68 | OOM |
| SAMCL | 0.9384 ± 0.66 | 0.9501 ± 0.85 | 0.9353 ± 0.46 | 0.8564 ± 1.61 | 0.8561 ± 0.45 | OOM | 0.5857 ± 0.86 | 0.4969 ± 1.59 |
| HUGE | 0.9537 ± 0.19 | 0.9693 ± 0.17 | 0.9485 ± 0.09 | 0.9046 ± 0.37 | 0.9049 ± 0.17 | OOM | 0.5906 ± 0.39 | 0.6015 ± 0.42 |
| **SCALER** | **0.9666 ± 0.35** | **0.9804 ± 0.38** | **0.9621 ± 0.08** | **0.9557 ± 0.24** | **0.9326 ± 0.27** | **0.9050 ± 0.13** | 0.5937 ± 0.08 | **0.6211 ± 0.26** |

Table 8: Complete results (mean ± std(%)) with AUPRC. The best and second-best results are in **bold** and underlined, respectively. OOM indicates out-of-memory on a 16GB GPU.

| Method | Cora | Citeseer | ACM | Citation | DBLP | ogbn-Arxiv | Reddit | YelpChi |
|---|---|---|---|---|---|---|---|---|
| DOMINANT | 0.4013 ± 1.16 | 0.2902 ± 1.08 | 0.1729 ± 0.00 | 0.3693 ± 0.05 | 0.4356 ± 0.07 | OOM | 0.0370 ± 0.00 | OOM |
| CoLA | 0.4141 ± 5.62 | 0.3882 ± 1.44 | 0.3085 ± 0.28 | 0.2897 ± 1.33 | 0.2826 ± 2.10 | OOM | 0.0409 ± 0.11 | 0.0421 ± 0.08 |
| ANEMONE | 0.4472 ± 3.41 | 0.4784 ± 3.51 | 0.3355 ± 0.52 | 0.2537 ± 2.92 | 0.3023 ± 1.31 | OOM | 0.0417 ± 0.06 | 0.0420 ± 0.08 |
| SL-GAD | 0.5323 ± 1.17 | 0.5064 ± 3.34 | 0.2995 ± 0.97 | 0.3777 ± 1.05 | 0.4172 ± 0.34 | OOM | 0.0369 ± 0.09 | 0.0432 ± 0.08 |
| Sub-CR | 0.5453 ± 6.60 | 0.5544 ± 2.46 | 0.3058 ± 1.05 | 0.2992 ± 2.12 | 0.2976 ± 5.81 | OOM | 0.0370 ± 0.48 | 0.0443 ± 1.00 |
| GRADATE | 0.5189 ± 2.62 | 0.2660 ± 1.08 | 0.3161 ± 1.69 | 0.1764 ± 1.37 | 0.2828 ± 1.05 | OOM | 0.0423 ± 0.26 | OOM |
| NLGAD | 0.3906 ± 0.91 | 0.4970 ± 3.47 | 0.2766 ± 0.26 | 0.1996 ± 1.63 | 0.2919 ± 2.64 | OOM | 0.0399 ± 0.06 | 0.0434 ± 0.06 |
| ADA-GAD | 0.3387 ± 0.18 | 0.3443 ± 0.18 | OOM | 0.3203 ± 0.01 | 0.3543 ± 0.08 | OOM | 0.0406 ± 0.00 | OOM |
| GADAM | 0.7577 ± 1.58 | 0.7096 ± 1.22 | 0.4432 ± 0.23 | 0.3778 ± 1.65 | 0.4175 ± 1.53 | 0.2349 ± 0.94 | 0.0485 ± 0.16 | 0.0633 ± 0.63 |
| DiffGAD | 0.3867 ± 0.02 | 0.3237 ± 0.01 | OOM | 0.3782 ± 0.00 | 0.4356 ± 0.01 | OOM | 0.0371 ± 0.02 | 0.0659 ± 0.34 |
| AD-GCL | 0.4933 ± 5.35 | 0.5590 ± 2.24 | 0.4026 ± 0.42 | 0.1932 ± 1.44 | 0.2124 ± 0.78 | OOM | 0.0394 ± 0.32 | OOM |
| SAMCL | 0.5838 ± 5.29 | 0.5916 ± 6.07 | 0.3519 ± 2.79 | 0.3422 ± 1.43 | 0.3343 ± 0.34 | OOM | 0.0395 ± 0.13 | 0.0554 ± 0.14 |
| HUGE | 0.7056 ± 1.48 | 0.7780 ± 0.48 | 0.4623 ± 0.25 | 0.4804 ± 2.42 | 0.5345 ± 1.93 | OOM | 0.0509 ± 0.20 | 0.0718 ± 0.17 |
| **SCALER** | **0.8307 ± 1.16** | **0.7918 ± 1.49** | **0.5035 ± 0.45** | **0.6437 ± 2.16** | **0.6646 ± 1.17** | **0.3108 ± 0.31** | **0.0523 ± 0.17** | **0.0806 ± 0.15** |

### E.3 HYPERPARAMETER SENSITIVITY ANALYSIS

In this section, we analyze the effects of several hyperparameters: $\alpha$, $\beta$, $\gamma$ and the number of negative edges. Considering comprehensive multi-metric performance and robustness, the detailed configurations of hyperparameters are reported in Appendix D.5.

#### E.3.1 THE INFLUENCE OF HYPERPARAMETER $\alpha$

We first study the impact of the hyperparameter $\alpha$, which controls the contribution of node-level contrast in the MLP embedding space, under the **w/ node contrast** variant. As shown in Fig. 7, the optimal performance is achieved at $\alpha = 1.0$ for YelpChi, $\alpha = 0.4$ for Citeseer, and $\alpha = 0.1$ for the other datasets. Incorporating node-level contrast in the GNN embedding space enables the MLP to better learn structural information and local relational patterns. Meanwhile, assigning a moderate weight to the loss in the MLP embedding space helps maintain feature uniqueness and stabilize feature distributions. For most datasets, smaller weights ($\alpha = 0.1$ or $\alpha = 0.4$) yield better performance because excessively large weights introduce redundant information that diminishes structural information gains, as GNN embeddings are learned based on MLP embeddings. Interestingly, YelpChi

Table 9: Complete results (mean ± std(%)) with Recall@K. The best and second-best results are in **bold** and underlined, respectively. OOM indicates out-of-memory on a 16GB GPU.

| Method | Cora | Citeseer | ACM | Citation | DBLP | ogbn-Arxiv | Reddit | YelpChi |
|---|---|---|---|---|---|---|---|---|
| DOMINANT | $0.4750 \pm 2.33$ | $0.4067 \pm 1.25$ | $0.2083 \pm 0.00$ | $0.4556 \pm 0.16$ | $0.4908 \pm 0.55$ | OOM | $0.0082 \pm 0.00$ | OOM |
| CoLA | $0.4283 \pm 3.84$ | $0.4250 \pm 0.87$ | $0.4022 \pm 0.80$ | $0.3117 \pm 1.77$ | $0.3050 \pm 1.72$ | OOM | $0.0317 \pm 0.48$ | $0.0394 \pm 0.12$ |
| ANEMONE | $0.4517 \pm 1.91$ | $0.4867 \pm 3.56$ | $0.4246 \pm 0.41$ | $0.2800 \pm 3.04$ | $0.3358 \pm 0.28$ | OOM | $0.0451 \pm 0.27$ | $0.0403 \pm 0.12$ |
| SL-GAD | $0.5067 \pm 1.56$ | $0.4867 \pm 3.50$ | $0.4008 \pm 1.35$ | $0.4111 \pm 2.10$ | $0.4308 \pm 1.11$ | OOM | $0.0314 \pm 0.24$ | $0.0485 \pm 0.30$ |
| Sub-CR | $0.5300 \pm 5.55$ | $0.5367 \pm 2.24$ | $0.4121 \pm 1.05$ | $0.3267 \pm 0.87$ | $0.3425 \pm 3.48$ | OOM | $0.0300 \pm 1.59$ | $0.0374 \pm 1.74$ |
| GRADATE | $0.5033 \pm 2.77$ | $0.3517 \pm 2.13$ | $0.3842 \pm 2.64$ | $0.2028 \pm 0.81$ | $0.3278 \pm 1.29$ | OOM | $0.0355 \pm 0.58$ | OOM |
| NLGAD | $0.4533 \pm 0.33$ | $0.5400 \pm 3.00$ | $0.3183 \pm 0.68$ | $0.2688 \pm 0.82$ | $0.3266 \pm 1.55$ | OOM | $0.0410 \pm 0.35$ | $0.0323 \pm 0.10$ |
| ADA-GAD | $0.5000 \pm 0.00$ | $0.5133 \pm 0.00$ | OOM | $0.3674 \pm 0.21$ | $0.3922 \pm 0.16$ | OOM | $0.0437 \pm 0.22$ | OOM |
| GADAM | $\underline{0.7120 \pm 1.36}$ | $\underline{0.7120 \pm 1.36}$ | $0.4900 \pm 0.71$ | $0.4840 \pm 1.97$ | $\underline{0.6113 \pm 2.83}$ | $\underline{0.2627 \pm 1.44}$ | $0.0743 \pm 0.84$ | $0.0896 \pm 1.22$ |
| DiffGAD | $0.5000 \pm 0.00$ | $0.3800 \pm 0.00$ | OOM | $0.4689 \pm 0.00$ | $0.4900 \pm 0.00$ | OOM | $0.0082 \pm 0.00$ | $0.0862 \pm 1.30$ |
| AD-GCL | $0.4883 \pm 3.00$ | $0.6417 \pm 1.66$ | $0.4739 \pm 1.16$ | $0.2567 \pm 0.68$ | $0.2758 \pm 1.30$ | OOM | $0.0335 \pm 1.50$ | OOM |
| SAMCL | $0.5733 \pm 3.40$ | $0.5800 \pm 4.37$ | $0.4378 \pm 3.00$ | $0.3726 \pm 1.00$ | $0.3722 \pm 0.63$ | OOM | $0.0273 \pm 0.59$ | $0.0674 \pm 0.24$ |
| HUGE | $0.6467 \pm 0.73$ | $0.6840 \pm 1.08$ | $\underline{0.5443 \pm 0.62}$ | $\underline{0.5018 \pm 2.79}$ | $0.5527 \pm 1.90$ | OOM | $\underline{0.0847 \pm 0.57}$ | $\underline{0.0989 \pm 0.96}$ |
| **SCALER** | $\mathbf{0.7533 \pm 1.19}$ | $\mathbf{0.7640 \pm 0.68}$ | $\mathbf{0.5780 \pm 1.25}$ | $\mathbf{0.6391 \pm 2.22}$ | $\mathbf{0.6473 \pm 0.39}$ | $\mathbf{0.3470 \pm 0.43}$ | $\mathbf{0.0956 \pm 0.85}$ | $\mathbf{0.1136 \pm 0.68}$ |

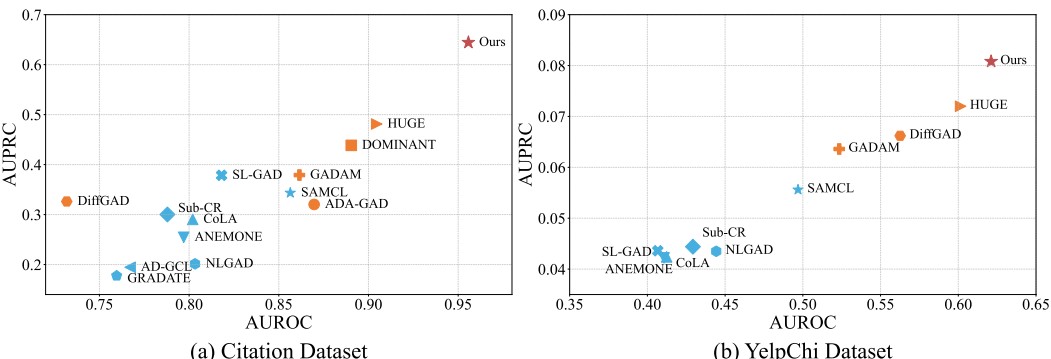

(a) Citation Dataset      (b) YelpChi Dataset

Figure 5: Performance comparison of various methods in terms of AUROC and AUPRC on the (a) Citation and (b) YelpChi datasets, where our method achieves superior results.

performs better with larger $\alpha$, indicating that attribute information plays a more critical role than structural information for this dataset.

### E.3.2 THE INFLUENCE OF HYPERPARAMETER $\beta$

We use the hyperparameter $\beta$ to control the weight of the edge-level contrastive loss. Due to the scale differences with the node-level contrastive loss, $\beta$ is varied over $\{0.1, 0.5, 1.0, 2.0, 4.0\}$. The results in Fig. 8 show that datasets exhibit varying sensitivities to edge-level contrast information. For Cora, Citeseer, DBLP, Citation, and Reddit, the AUROC score initially increases and then decreases as $\beta$ grows. This indicates that a moderate amount of edge-level contrastive information enables the model to effectively leverage edge information beyond node features, thereby improving anomaly detection performance. Interestingly, performance on the ACM dataset continues to improve with higher $\beta$, suggesting that graph structural information in ACM is underexploited. The additional signals introduced by edge-level contrast compensate for this underutilization. The trend on YelpChi is opposite: performance declines as $\beta$ increases. This observation aligns with earlier analysis, where node attributes in YelpChi play a more dominant role than structural information.

### E.3.3 THE INFLUENCE OF HYPERPARAMETER $\gamma$

We use the hyperparameter $\gamma$ to control the weight of the regularization term, which mitigates potential distribution shifts. Since the magnitude of this term is relatively small, we scale $\gamma$ by $10^4$

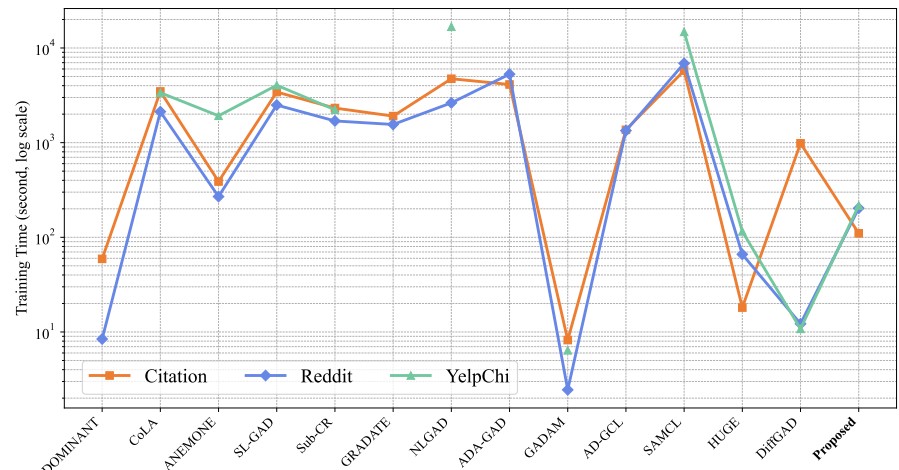

Figure 6: Training time comparison across different methods.

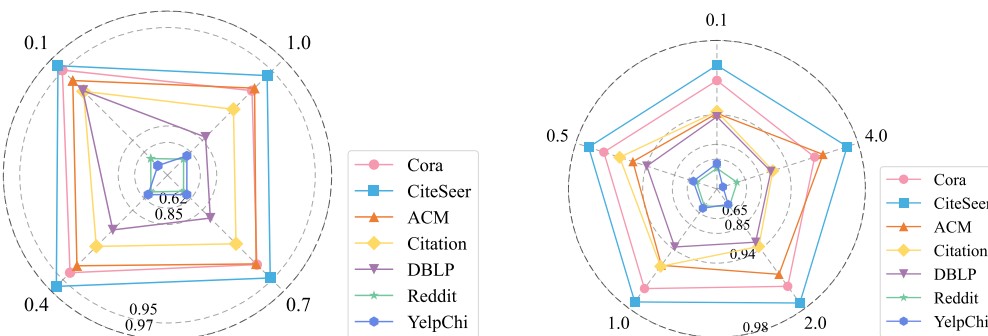

Figure 7: Sensitivity analysis of hyperparameter $\alpha$ w.r.t. AUROC.

Figure 8: Sensitivity analysis of hyperparameter $\beta$ w.r.t. AUROC.

to ensure its effect is significant. In our experiments, $\gamma$ is selected from $\{0, 0.5, 1.0, 2.0, 4.0\}$. The results in Fig. 9 show that the sensitivity to this parameter differs across datasets. Compared with the unregularized version (i.e., $\gamma = 0$), most datasets achieve performance improvements. For non-large-scale datasets, increasing the edge-level contrastive loss weight generally necessitates a higher $\gamma$. This finding suggests that, although the edge-level contrastive strategy effectively incorporates edge-level information into the contrastive learning network, it may also shift the feature distribution of the original embedding space. By jointly tuning both weights, we can appropriately introduce edge-level contrast, thereby improving anomaly detection performance. However, for the two datasets with organic anomalies, Reddit and YelpChi, the best performance is achieved when $\gamma = 0$. This is mainly because the distribution shift introduced by self-supervised training is relatively weak, and excessive regularization disrupts the original feature distribution, thereby reducing detection performance.

### E.3.4 THE SENSITIVITY OF THE NUMBER OF NEGATIVE EDGES

To reduce computational overhead, we construct a global negative edge pool through sampling and adopt simple random sampling to keep the process efficient. Fig. 10 shows AUROC performance on the Citation, DBLP, and YelpChi datasets as the size of the negative edge pool varies. We observe that, due to the randomness of simple random sampling, performance fluctuates at small pool sizes. However, as the pool size increases, performance stabilizes with only minor changes. Overall, the edge-level contrastive learning strategy is relatively insensitive to the number of negative edges. Therefore, we fixed this number at 60 in our experiments.

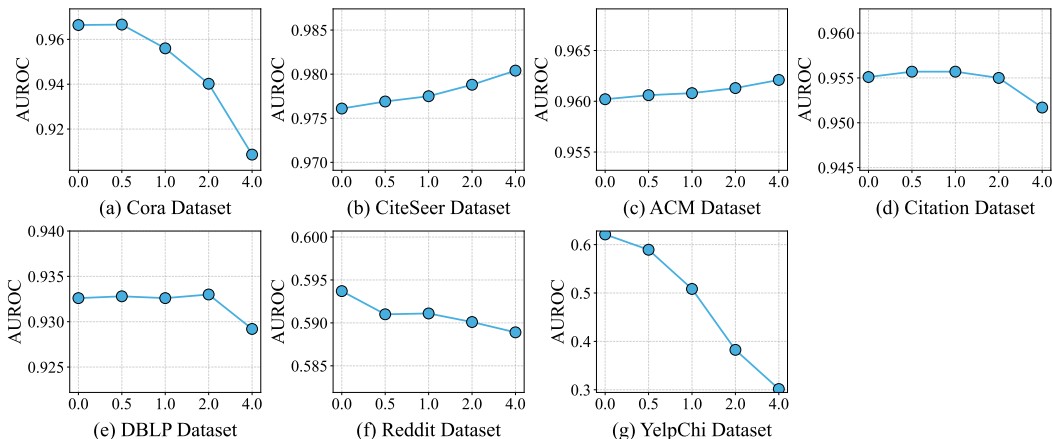

Figure 9: Sensitivity analysis of regularization hyperparameter $\gamma$ (x$10^4$) w.r.t. AUROC.

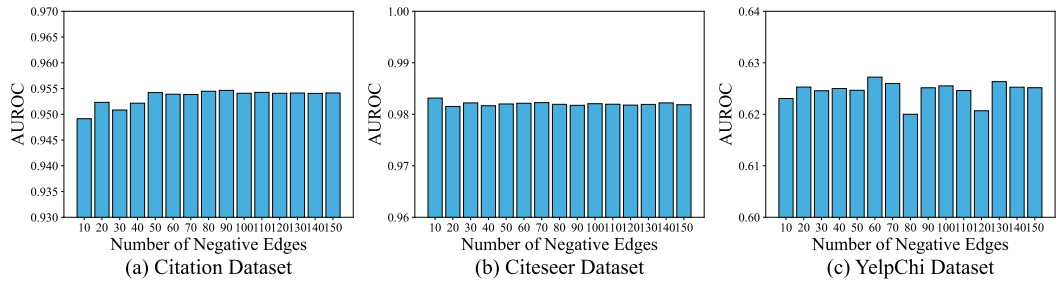

Figure 10: Sensitivity analysis of the number of negative edges on the (a) Citation, (b) Citeseer, and (c) YelpChi datasets w.r.t. AUROC.

In contrastive learning, selection of positive and negative samples is critical. In proposed edge-level contrastive strategy, negative edges for each edge are all edges that do not share endpoints with it. Since the number of sampled negative edges is much smaller than the total number of edges, the probability that the global negative pool contains a large fraction of positive edges is extremely low, ensuring that edge-level contrastive learning remain effective. Consequently, using simple random sampling to obtain global negative edges is reasonable.

### E.4 VISUALIZATION

#### E.4.1 VISUALIZATION OF THE ANOMALY LEARNING PROCESS DURING TRAINING

We compare the anomaly learning process under two frameworks: the Only-MLP framework and the MLP-GNN framework. Fig. 11 illustrates how the neighborhood similarity and anomaly score of the target node evolve over training epochs. In the Only-MLP setting, node-level contrast can indirectly incorporate some structural cues but fall short of fully capturing neighborhood information, leading to effective yet suboptimal detection. In contrast, the MLP-GNN framework, guided by dual-level contrastive network, integrates structural and neighborhood knowledge from the GNN into the MLP effectively, enabling it to better exploit local neighborhood signals, capture more anomalous patterns, and ultimately yield higher anomaly scores.

#### E.4.2 VISUALIZATION OF THE DISTRIBUTION OF ANOMALY SCORES WITH CORRECTION

When choosing the value range, the key principle is to preserve the distributional gap between normal and anomalous nodes; otherwise, the two classes may become mixed, which undermines the algorithm's detection capability. A relatively small range allows for fine-tuning the anomaly scores of certain low-entropy anomalous nodes, thereby improving performance. In our experiments, we set

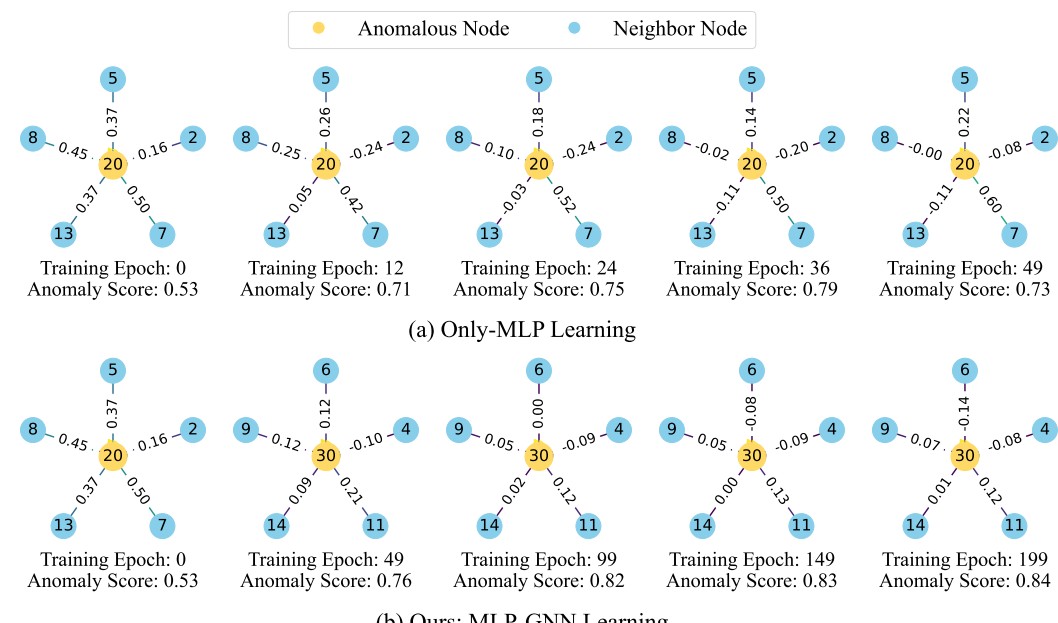

(a) Only-MLP Learning

(b) Ours: MLP-GNN Learning

Figure 11: Visualization of the anomaly learning process during training under (a) Only-MLP and (b) MLP-GNN learning framework. Node IDs are annotated above the nodes, cosine similarities are displayed on the edges, and the anomaly score of the anomalous node at the current training epoch is indicated.

the maximum value to 0.2. Fig. 12 illustrates the anomaly score distributions on the Cora and DBLP datasets before and after correction. The module does not substantially alter the overall anomaly scores and the score distribution gap between normal and anomalous nodes. However, it selectively adjusts the scores of certain low-entropy anomalous nodes, pushing them further away from normal nodes and closer to the anomalous nodes, thereby improving anomaly detection performance and enhancing robustness.

### E.5 GENERALIZATION ANALYSIS OF STRUCTURE-AWARE MLP

In this section, we investigate the anomaly detection performance of the trained structure-aware MLP on new subgraphs. New subgraphs can be divided into two categories:

- Independent subgraphs, which have no edge connections to the original graph (Fig. 13).
- Extended subgraphs, which remain connected to the original graph (Fig. 14).

### E.5.1 ANOMALY DETECTION PERFORMANCE ON INDEPENDENT SUBGRAPHS

To evaluate the structure-aware performance and anomaly detection capability of MLP trained under the MLP-GNN framework with a dual-level synergistic contrastive strategy, we first evaluated detection performance on newly introduced subgraphs, which are completely disconnected from the original graph. To identify anomalies within these subgraphs, three strategies are considered: (i) unsupervised training on the entire graph including the original graph and the new subgraph, (ii) training exclusively on the new subgraph, and (iii) transferring a model pre-trained on the original graph. We conducted experiments on YelpChi datasets. The entire graph was partitioned into two subgraphs, the original graph $\mathcal{G}_1$ and the new subgraph $\mathcal{G}_2$, based on connected components. The proposed algorithm was then applied to train the optimal MLP models separately on $\mathcal{G}_1 + \mathcal{G}_2$, $\mathcal{G}_2$, and $\mathcal{G}_1$, yielding three distinct models. Subsequently, anomaly estimation was performed on $\mathcal{G}_2$, and the AUROC scores were computed. To better evaluate performance under varying structural information, we considered three different ratios of the number of nodes between $\mathcal{G}_1$ and $\mathcal{G}_2$: (a) 1:1, (b) 2:1, and (c) 1:2. The experimental results are presented in Fig. 13. The results show that well-trained

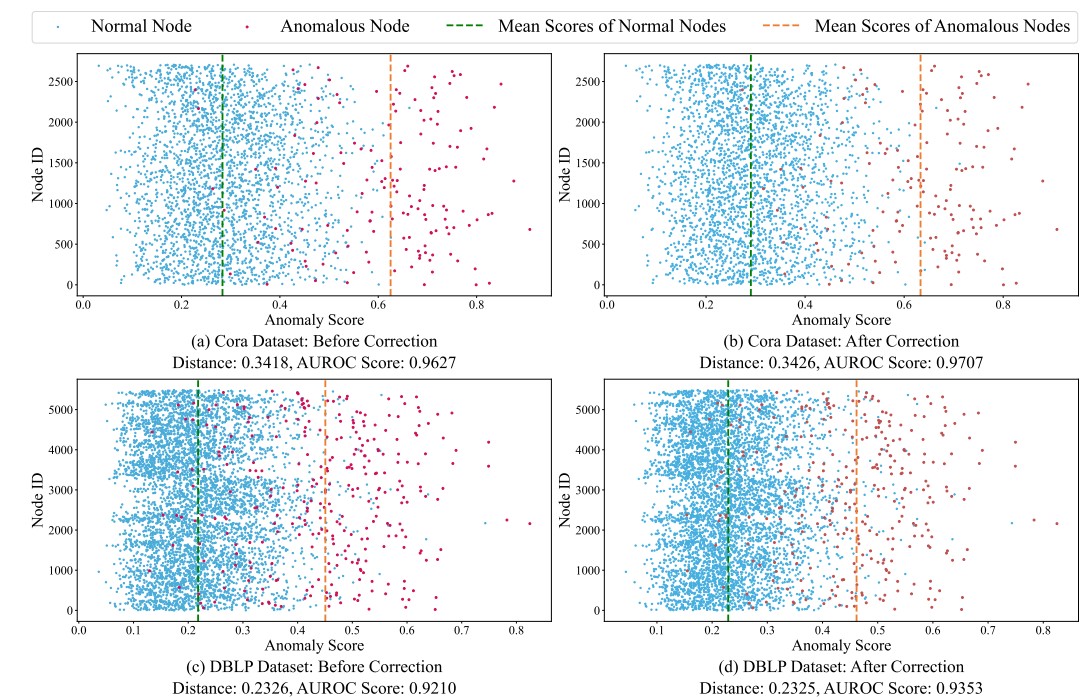

Figure 12: Visualization of the distribution of anomaly scores before and after correction on (a, b) Cora and (c, d) DBLP datasets. The figure also indicates the mean anomaly scores of normal and anomalous nodes, their distance, and the AUROC score.

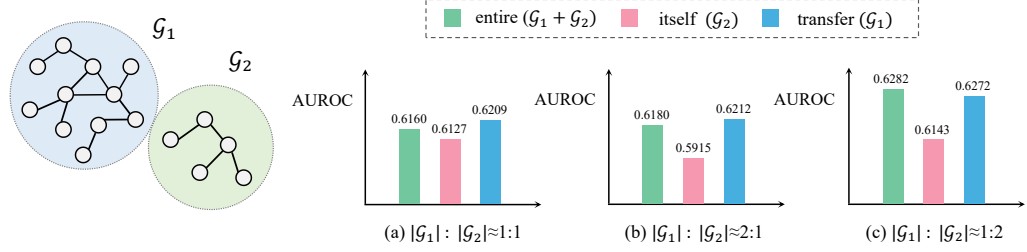

Figure 13: Anomaly detection performance on independent subgraphs of MLP under three different ratios of the number of nodes between $\mathcal{G}_1$ and $\mathcal{G}_2$: (a) 1:1, (b) 2:1, and (c) 1:2.

MLPs consistently achieved high performance on new subgraphs, in some cases even surpassing the performance obtained when trained on the full or original graph. This demonstrates that MLP trained using this algorithm possess strong structural awareness and maintain strong transferability even when the graph size is moderately imbalanced.

### E.5.2 ANOMALY DETECTION PERFORMANCE ON EXTENDED SUBGRAPHS

Next, we evaluated detection performance on subgraphs extended from the original graph, where the new subgraphs remain connected to the original structure. Experiments were conducted on the Reddit and ACM datasets. We simulate the extended-subgraph scenario by selecting the largest connected component and partitioning it into two subgraphs, $\mathcal{G}_1$ and $\mathcal{G}_2$. We use a 9:1 node ratio to partition to avoid drastic structural disruption. We then compare two training schemes: (a) Full-graph training: training the entire model on the complete graph and testing on $\mathcal{G}_2$, and (b) Partial-graph transfer: training only on $\mathcal{G}_1$ and directly transferring the trained MLP to $\mathcal{G}_2$ for testing. This setup evaluates whether the MLP can generalize its structure-aware representations to graph regions unseen during training. As illustrated in Fig. 14, the pre-trained MLP remains competitive compared

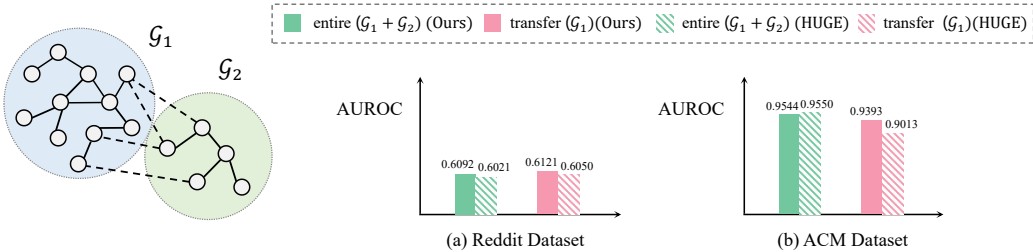

Figure 14: Anomaly detection performance on extended subgraphs of MLP for the (a) Reddit and (b) ACM datasets. Dashed curves indicate the extended subgraphs and their linkage to the original graph.

Table 10: AUROC comparison on two types of anomalies.

| Methods | Cora | | Citeseer | | ACM | | Citation | | DBLP | |
|---|---|---|---|---|---|---|---|---|---|---|
| | Contextual | Structural | Contextual | Structural | Contextual | Structural | Contextual | Structural | Contextual | Structural |
| **SCALER** | **0.9217** | **0.9929** | **0.9468** | **0.9953** | **0.9392** | **0.9693** | **0.9257** | **0.9698** | **0.8610** | **0.9851** |
| HUGE | 0.8950 | 0.9927 | 0.9208 | 0.9953 | 0.9129 | 0.9633 | 0.8152 | 0.9603 | 0.7997 | 0.9833 |
| GADAM | 0.8779 | 0.9907 | 0.8962 | 0.9934 | 0.9334 | 0.9577 | 0.7554 | 0.9525 | 0.7298 | 0.9846 |
| CoLA | 0.8306 | 0.9094 | 0.8087 | 0.9071 | 0.9224 | 0.8446 | 0.7865 | 0.8015 | 0.7368 | 0.8089 |

to unsupervised training on the entire graph. Compared with HUGE, this method jointly trains the MLP and GNN using globally heterogeneous-guided ranking and alignment losses. However, when transferring to the new subgraph, HUGE lacks a heterogeneity-aware metric for the new subgraph, leading to degraded structural perception and reduced anomaly detection performance. Our method achieves superior transferability. This conclusion is also supported by the strong performance that HUGE attains on the ACM dataset under full-graph training, further indicating that its performance collapses mainly during transfer. Overall, the pre-trained structure-aware MLP demonstrates strong generalization ability while effectively mitigating inference latency issues in real-world scenarios.

### E.6 CONTEXTUAL AND STRUCTURAL ANOMALY DETECTION PERFORMANCE ANALYSIS

We study the performance of different method on two types of injected anomalies. We select three representative methods in the baselines to form comparison experiments: CoLA, GADAM and HUGE. The results are recorded in Tab. 10. From the experimental results, we draw the following conclusions:

- Most methods outperform contextual anomaly detection for structural anomaly detection on five synthetic datasets, indicating that structural anomalies produce stronger anomaly signals and are therefore easier to detect. However, there remains substantial room for improvement in detecting contextual anomalies.

- Effective anomaly detection methods must balance performance across both types of anomalies. Our method achieves the best trade-off, demonstrating superior effectiveness in detecting both structural and contextual anomalies. Notably, by combining two complementary contrastive strategies to fully exploit the graph information, our method substantially enhances the detection performance on contextual anomalies.

## F NOTATIONS

To facilitate understanding, we summarize the primary mathematical notations used in this paper in Tab. 11.

Table 11: Summary of the primary notations.

| Symbol | Description | Symbol | Description |
|---|---|---|---|
| $\mathcal{G}$ | An attributed network | $\mathcal{V}$ | The node set of $\mathcal{G}$ |
| $v_i$ | The i-th node of $\mathcal{G}$ | $\mathcal{E}$ | The edge set of $\mathcal{G}$ |
| $\mathbf{X}$ | Node features matrix of $\mathcal{G}$ | $\mathbf{X}_i$ | The i-th row of $\mathbf{X}$ |
| $\mathbf{A}$ | A binary adjacency matrix | $\mathbf{D}$ | The degree matrix of $\mathcal{G}$ |
| $\mathbf{S}$ | The anomaly scores of all nodes | $\mathcal{S}^{\mathrm{MLP}}$ | The MLP embedding space |
| $\mathcal{S}^{\mathrm{GNN}}$ | The GNN embedding space | $\mathbf{H}^{\mathrm{MLP}}$ | The MLP embedding |
| $\mathbf{H}^{\mathrm{GNN}}$ | The GNN embedding | $\widetilde{\mathbf{D}}$ | The degree matrix with self-loops |
| $\widetilde{\mathbf{A}}$ | Adjacency matrix with self-loops | $\mathbf{I}$ | Identity matrix |
| $\mathbf{W}^{(\ell-1)}$ | Weight matrix of $(\ell-1)$-th GCN layer | $\mathcal{N}(v_i)$ | First-order neighbors of $v_i$ |
| $\mathcal{B}$ | Set of all nodes in this batch | $\mathcal{N}^+(v_i), \mathcal{N}^-(v_i)$ | Positive and negative neighborhoods of $v_i$ |
| $c_i^+, c_i^-$ | Consistency scores of $v_i$ | $\mathbf{H}_i^{\mathrm{GNN}}$ | Embedding of $v_i$ in $\mathcal{S}^{\mathrm{GNN}}$ |
| $e_{ij}$ | Target edge | $\mathbf{E}_{ij}^{\mathrm{GNN}}$ | Embedding of $e_{ij}$ in $\mathcal{S}^{\mathrm{GNN}}$ |
| $\mathcal{P}_{ij}^+$ | Positive edge pool of $e_{ij}$ | $\mathcal{P}_{ij}^{+,1}, \mathcal{P}_{ij}^{+,2}, \mathcal{P}_{ij}^{+,3}$ | Three positive sub-pools |
| $\mathcal{P}_{ij}^-$ | Negative edge pool of $e_{ij}$ | $\mathbf{z}_{ij}^k$ | Embedding of k-th positive sub-pool |
| $\tau$ | Temperature parameter | $s_i$ | Anomaly score of $v_i$ |
| $q_{ij}$ | Probability distribution within cosine similarity | $t_i, \tilde{t}_i$ | Neighborhood entropy and normalized neighborhood entropy of $v_i$ |