# OpenReview forum: "SCALER: Fast and Effective Graph Anomaly Detection via Dual-Level Synergistic Contrastive Learning"
_ICLR.cc/2026/Conference — Submitted to ICLR 2026_

### Official Review · Reviewer_pyNh · 2025-10-26

**Soundness:** 3
**Presentation:** 3
**Contribution:** 2
**Rating:** 4
**Confidence:** 3

**Summary:**

This paper addresses the challenge of UGAD, where existing deep learning methods suffer from long inference times due to neighborhood aggregation and limitations in contrastive learning strategies. The authors propose SCALER, a self-supervised MLP-GNN framework that trains a structure-aware MLP through dual-level synergistic contrastive learning. The method incorporates node-level and edge-level contrastive networks to capture structural patterns, and introduces a neighborhood entropy-guided anomaly score correction module.

**Strengths:**

1. Using only MLP for inference is an interesting approach, and experimental results show that it requires only a small amount of runtime.
2. The paper designs an anomaly score correction module based on neighbor entropy, optimizing for low-entropy anomaly nodes and avoiding the limitations of traditional methods that rely solely on consistency scores.
3. The experimental design is detailed and comprehensive, and the results are excellent.

**Weaknesses:**

1. Multi-scale graph contrastive learning has achieved excellent results, and the differences between dual-level contrastive learning and existing methods are worth discussing.
2. How should we understand "dual-level synergistic"? It seems to be just a simple combination of different perspectives.
3. Dual-level contrastive learning is all about the design of low-order neighbors. If the graph contains long-range dependencies, MLP may not perform well.

**Questions:**

Have the authors considered contrastive learning at the sub-graph level?

---

> ### Author Response · Authors · 2025-11-20
> **Response to Reviewer pyNh ---- Part 1/2**
>
> We sincerely appreciate your insightful comments and acknowledgment of our contributions. Below, we provide point-to-point responses to address your concerns and clarify the confusion of our proposed method.
> > **W1.** The differences between dual-level contrastive learning and existing methods.
>
> While multi-scale graph contrastive learning has achieved excellent results, **existing GCL strategies in UGAD primarily focus on node-level and subgraph-level contrast**. CoLA, the first contrastive-based UGAD method, introduced the node–subgraph contrast paradigm, and most subsequent works follow and refine this paradigm. For example, ANEMONE integrates node–node contrast into the node–subgraph framework. GRADATE utilizes a multi-scale, multi-view contrastive framework and performs subgraph-level alignment, while SAMCL conducts the interview subgraph-aligned contrastive learning module to better detect changes for nodes with different local subgraphs. More details on contrastive–based UGAD methods can be found in Appendix B  RELATED WORK(page 12-13 of the paper).
>
> **However, none of these methods explicitly incorporate edge-level contrast. Our proposed dual-level contrastive network fills this gap by introducing egde-level contrast for GAD tasks.** Some GAD methods, such as BOURNE, have also highlighted the insufficient exploitation of edge-level information. However, these approaches typically enhance edge-related signals indirectly by employing hypergraph transformations or constructing multiple structural views, rather than explicitly designing a contrastive objective at the edge level. In contrast, our method directly incorporates edge-level contrast and integrates it with node-level contrast, thereby enabling more effective modeling of complex structural patterns.
>
> In summary, unlike prior multi-scale contrastive learning methods in GAD, **our dual-level contrastive network provides a novel and complementary edge-level perspective**, enabling the model to capture richer and more complex anomaly patterns and thereby improving anomaly detection performance.
>
> > **W2.** How should we understand "dual-level synergistic"? It seems to be just a simple combination of different perspectives.
>
> During training, our dual-level contrastive network effectively captures multi-scale structural information and guides the MLP to learn structure-aware representations. By "dual-level synergistic," we do not mean a simple combination of objectives. **Instead, the node-level and edge-level objectives mutually reinforce each other throughout training. Unlike straightforward multi-view combination methods, our method leverages a distillation-based design in which a dual-level contrastive network synergistically distills structural information into the MLP.**
>
> Specifically, node-level contrast models the relationship between a node and its neighborhood structure, while edge-level contrast takes a more granular edge-level perspective, providing richer edge and structural information for node representation learning. These complementary perspectives make anomalous patterns more distinguishable. Both objectives operate on the same embedding space, and their gradients jointly shape the node representation, producing more discriminative and structure-aware features than any single-level objective alone. As shown in the ablation results in Table 2 (page 8 of the paper), the dual-level contrastive network achieves the best overall performance.

---

> > ### Comment · Reviewer_pyNh · 2025-11-27
> >
> > Thanks for the replies. These discussions have made this paper much clearer. After reading the comments from other reviewers, I decided to keep my score unchanged.

---

> ### Author Response · Authors · 2025-11-20
> **Response to Reviewer pyNh ---- Part 2/2**
>
> > **W3.** Dual-level contrastive learning is all about the design of low-order neighbors. If the graph contains long-range dependencies, MLP may not perform well.
>
> Thanks for your thoughtful comment and question.
>
> **In UGAD, the success of contrastive-based methods largely stems from modeling relationships between nodes and their local structural patterns, as exemplified by the node–subgraph contrast paradigm in CoLA.** Many subsequent works (e.g., ANEMONE, SL-GAD, Sub-CR, GRADATE, NLGAD, AD-GCL, SAMCL, GADAM) follow this paradigm. Moreover, for most benchmark in GAD datasets, without higher-order structural information, our experiment results further demonstrate that the structure-aware MLP learned by our dual-level contrastive network can achieve strong performance and surpass current state-of-the-art UGAD methods.
>
> You raised a valid concern that MLPs may underperform on graphs with pronounced long-range dependencies. Indeed, some long-range graph benchmark (LRGB) datasets, such as PascalVOC-SP, COCO-SP, PCQM-Contact, Peptides-func, and Peptides-struct, require reasoning over long-range interaction (LRI) to achieve competitive results. **However, the anomaly detection datasets we use exhibit weak or negligible long-range dependencies, so using an MLP does not degrade performance.** That said, we agree that current UGAD frameworks seldom consider scenarios where long-range dependencies play a central role. While low-order neighbor-based contrastive objectives may underutilize long-range information, the framework can be further strengthened by explicitly improving the extraction of long-range dependency signals into the MLP.
>
> To address the shortcomings of GNNs in capturing long-range dependencies, several recent works have explored incorporating global structural information. Graph Transformers (GTs) [1–4] have emerged as a powerful class of models for graph representation learning. By leveraging positional encodings and global self-attention, GTs can capture long-range dependencies and alleviate oversmoothing issues that commonly affect GNNs. However, their self-attention operations and architectural complexity incur substantial computational and memory costs, limiting their practical deployment. Recently, knowledge distillation (KD) methods [5] have been proposed to transfer the rich representations learned by a GT teacher to a lightweight MLP student. **These findings suggest that MLPs can remain effective even on graphs with long-range dependencies, provided that long-range structural knowledge is properly distilled into the MLPs.**
>
> > **Q1.** Have the authors considered contrastive learning at the sub-graph level?
>
> Thank you for raising this point.
>
> **Several recent UGAD works have indeed explored subgraph-level contrastive learning.** For example, SAMCL is the first approach to incorporate subgraph–subgraph level contrastive learning over existing GCL on GAD task. It follows the framework of CoLA and deploys a novel subgraph-aligned contrastive strategy that measures subgraph pairwise similarity with Earth mover's distance (EMD), considering both embedding distribution and topology distance.
>
> However, SAMCL still follows the node–subgraph contrastive framework of CoLA, so incorporating subgraph-level contrast is naturally compatible with this framework and contributes to strong performance. In contrast, our framework is neighborhood-based rather than subgraph-based. Neighborhood sizes of all nodes are typically non-uniform, and subgraph sampling would introduce additional computational overhead, which is inconsistent with our motivation of providing fast inference. Therefore, subgraph-level contrast was not considered in the design of current dual-level contrastive network.
>
> **Nevertheless, extending our dual-level framework with subgraph-level contrast is an interesting direction for future exploration and may lead to richer node representations.** We appreciate you for highlighting this point and will consider it in future work.
>
> **Notes**
>
> All referenced baseline methods, including CoLA, ANEMONE, SL-GAD, Sub-CR, GRADATE, NLGAD, AD-GCL, SAMCL, and GADAM, are listed in the Experimental Settings on page 6 of the paper (Section 4.1).
>
> **References**
>
> [1] Dexiong Chen, et al. Structure-Aware Transformer for Graph Representation Learning. In ICML 2022.
>
> [2] Ladislav Rampášek, et al. Recipe for a General, Powerful, Scalable Graph Transformer. In NIPS 2022.
>
> [3] Yujie Xing, et al. Less is More: on the Over-Globalizing Problem in Graph Transformers. In ICML 2024.
>
> [4] Mitchell Black, et al. Comparing graph transformers via positional encodings. In ICML 2024.
>
> [5] Sarthak Malik, et al. GraTeD-MLP: Efficient Node Classification via Graph Transformer Distillation to MLP. In LoG 2024.
>
> Thank you again for your valuable comments and suggestions. If you have any further questions, we would be pleased to discuss them with you.

---

### Official Review · Reviewer_ndWk · 2025-10-31

**Soundness:** 3
**Presentation:** 3
**Contribution:** 3
**Rating:** 6
**Confidence:** 4

**Summary:**

This paper addresses the challenge of Unsupervised Graph Anomaly Detection (UGAD), particularly the high inference latency of existing deep learning methods due to neighborhood aggregation and limitations in current contrastive learning strategies, such as node-subgraph contrast. The authors propose SCALER, a self-supervised MLP-GNN learning framework that trains a structure-aware Multilayer Perceptron (MLP) for fast anomaly detection without requiring costly Graph Neural Network (GNN) aggregation during inference. Extensive experiments on eight benchmark datasets show that SCALER significantly improves detection performance (e.g., average gain of $19.6\%$ in AUPRC) while reducing inference time to the order of seconds, demonstrating high effectiveness and efficiency.

**Strengths:**

* High Efficiency with Performance Retention: The MLP-GNN framework successfully decouples costly GNN aggregation from the inference phase, leading to orders of magnitude faster anomaly estimation while achieving superior detection performance, a key practical advantage.
* Novel Dual-Level Contrastive Learning: The introduction of a more robust node-level neighborhood contrast (using first-order neighbors as positive samples) and a novel edge-level contrast effectively addresses the limitations of previous contrastive UGAD methods like sampling-based node-subgraph contrast.
* Strong Empirical Results: The method achieves state-of-the-art results across various metrics (AUROC, AUPRC, Recall@K) and datasets, demonstrating robust and generalizable detection capability. The performance is notably strong in the challenging AUPRC metric.
Scalability: The framework's ability to handle the large-scale ogbn-Arxiv dataset, where many baselines encounter out-of-memory issues, confirms its practical scalability.

**Weaknesses:**

* Complexity of Training Objective: The joint training objective involves three main loss terms (\mathcal{L}_{node}, \mathcal{L}_{edge},  \mathcal{L}_{reg})  controlled by three hyperparameters (\alpha, \beta, \gamma). Although a hyperparameter analysis is provided, the need to carefully tune three trade-off parameters adds complexity to deployment and reproduction on new datasets.
* Generalization Analysis Clarity: While the generalization analysis in Appendix E.5 is valuable, its description, especially the “Anomaly detection performance on extended subgraphs” section (E.5.2), could be clearer about the specific experimental setup and the connection between $G_1$ and $G_2$ (the dashed lines in Figure 13 are only present in the first diagram in the appendix, not the second that is referenced). The performance of the MLP trained only on $G_1$ on the unseen $G_2$ subgraph should be a strong point, but the comparison with HUGE is a bit obscured in the text

**Questions:**

* AUPRC vs. AUROC Trade-off on Reddit: On the Reddit dataset, SCALER’s AUROC is slightly below the best baseline (0.5937 vs. 0.5980 for GRADATE), but its AUPRC is the best (0.0523 vs. 0.0509 for HUGE). Given AUPRC is generally a better indicator for imbalanced anomaly detection, could the authors provide a more detailed interpretation of this slight discrepancy in AUROC, and how the dual-level contrastive strategy specifically contributes to the higher AUPRC and Recall@K on this dataset?
* Necessity of $\mathcal{L}_{reg}$ for Organic Anomalies: For the organic anomaly datasets Reddit and YelpChi, the best performance is achieved when the regularization weight $\gamma=0$. This suggests that for these graphs, the distribution shift is weak, and the regularization term is unnecessary or even detrimental. Could the authors expand on why the self-supervised training on these organic anomaly datasets introduces a weaker distribution shift compared to the synthetic anomaly datasets?
* Edge Feature Utilization: The edge embedding is simply defined as the average of its two endpoint node embeddings $E_{ij}^{GNN}=\frac{H_i^{GNN}+H_j^{GNN}}{2}$. For richer relational information, have the authors considered a more complex edge embedding function that might involve edge attributes (if available) or a dedicated edge feature learning module, and how might this impact overall performance and efficiency?

---

> ### Author Response · Authors · 2025-11-20
> **Response to Reviewer ndWk ---- Part 1/2**
>
> We highly appreciate your insightful comments and thoughtful feedback. Your constructive criticism is invaluable for refining our work.
> > **Q1.** AUPRC vs. AUROC Trade-off on Reddit.
>
> Thank you for the careful observation.
>
> All the graph anomaly detection datasets in our experiments are highly imbalanced, as they contain far fewer anomalous nodes than normal nodes. Among these datasets, **Reddit has the lowest anomaly ratio, making the imbalance particularly severe.** Under such conditions, AUROC can be misleading: a model may achieve a relatively high AUROC by correctly ranking many true negatives, while still fail to rank the very few anomalies at the top. This phenomenon is common in imbalanced anomaly detection. **It is therefore not surprising to see inconsistencies between AUROC and AUPRC across different methods on Reddit.**
>
> **AUPRC and Recall@K are more reliable metrics in this setting because they directly evaluate a model's ability to prioritize rare anomalies.** On Reddit, SCALER achieves the highest AUPRC and Recall@K, indicating that more true anomalies receive high anomaly scores early in the ranking. This improvement stems from the dual-level contrastive design:
> - **Node-level contrast** enhances the ability to separate abnormal nodes from their local neighborhoods.
> - **Edge-level contrast** provide a more granular edge-centric perspective by utilizing edge information to uncover abnormal connection patterns between nodes, thereby making it more effective at identifying anomalous node attributes.
>
> **These complementary views make anomalous patterns more distinguishable and improve the ranking quality.**
>
> Overall, the slight difference in AUROC is expected under such extreme imbalance, whereas the superior AUPRC and Recall@K more accurately reflect the effectiveness of SCALER's dual-level contrastive strategy on Reddit.
>
> > **Q2.** Necessity of $\mathcal{L}_{\text{reg}}$ for Organic Anomalies.
>
> **The observation is correct and can be explained by the intrinsic differences between organic and synthetic anomalies.** Details of the datasets and anomaly injection methods are provided in Appendix C (page 13-14 of the paper).
>
> In organic anomaly datasets, anomalous nodes arise naturally from real-world behaviors and often retain partial structural or attribute consistency with their surrounding neighborhoods. In contrast, synthetic anomaly datasets generate anomalies by artificially perturbing node attributes or rewiring edges, which creates clear and substantial distribution gaps between normal and anomalous samples. This phenomenon is even more pronounced in citation networks, where node features are typically high-dimensional and sparse. As a result, the distribution shift induced during self-supervised training is much weaker and more gradual in organic anomaly datasets. Introducing a regularization term that enforces stronger structural separation may therefore incorrectly push organically abnormal nodes away from their naturally coherent neighborhoods, leading to degraded performance. However, in synthetic anomaly datasets, where the distribution shift is large and abrupt, the regularization term effectively mitigates this bias and consequently enhances anomaly detection performance.
>
> > **Q3.** Edge Feature Utilization.
>
> In our method, edge embedding is defined as the average of its two endpoint node embeddings, chosen for efficiency during training. We agree that richer relational information can be incorporated through more sophisticated edge embedding functions. Therefore, we experimented with several alternative edge representation methods, and the AUROC results are shown in the table below.
>
> | Edge embedding function | Description | Cora | Citation |
> | -------- | ----    | ------ | ------  |
> | **Avg (Ours)**     | $\frac{\mathbf{H}^{\text{GNN}}_i + \mathbf{H}^{\text{GNN}}_j}{2}$   | 0.9666 | **0.9557**  |
> | Global Learnable Parameter         | $\lambda \mathbf{H}^{\text{GNN}}_i + (1 - \lambda) \mathbf{H}^{\text{GNN}}_j$   | **0.9667** | 0.9554  |
> | Adaptive Parameter  | $w _ {ij}^{(i)} \mathbf{H}^{\text{GNN}} _ i + w _ {ij}^{(j)} \mathbf{H}^{\text{GNN}} _ j  \quad and  \quad w _ {ij}^{(i)}, w _ {ij}^{(j)} = g([\mathbf{H}^{\text{GNN}} _ i \Vert \mathbf{H}^{\text{GNN}} _ j])$ | 0.9654 | 0.9504  |
> | Linear Transformation         | $\mathbf{W} \cdot \frac{\mathbf{H}^{\text{GNN}}_i + \mathbf{H}^{\text{GNN}}_j}{2}$   | 0.9518 | 0.9414  |
> | MLP Encoder         | $\mathbf{MLP}[\mathbf{H}^{\text{GNN}}_i \Vert \mathbf{H}^{\text{GNN}}_j]$   | 0.9579 | 0.9395  |
>
> **The results show that using more complex edge representation methods does not yield significant performance improvements; instead, it increases model complexity and training time.** In conclusion, our simple average edge representation strategy offers the best balance between efficiency and performance.

---

> ### Author Response · Authors · 2025-11-20
> **Response to Reviewer ndWk ---- Part 2/2**
>
> > **W1.** Complexity of Training Objective.
>
> We acknowledge that the joint training objective involves three loss terms associated with three trade-off hyperparameters, and each component supervises a different aspect of structural learning. Therefore, separate weights are required to balance their contributions.
> Although this introduces multiple hyperparameters, our sensitivity analysis shows that the model achieve strong performance within small ranges for each parameter. In fact, better results can be obtained by choosing values within the ranges provided in Table 5 (page 17 of the paper) rather than relying on finely tuned combinations.
>
> > **W2.** Generalization Analysis Clarity.
>
> Thanks for the careful reading and helpful comments.
>
> We agree that the description of the generalization analysis in Appendix E.5 could be clarified further. **We will revise this part (page 22-24 of the paper) to make the experimental setup and comparison more explicit.**
>
> In this section, we investigate the anomaly detection performance of the trained structure-aware MLP on new subgraphs. New subgraphs can be divided into two categories:
> - **Independent subgraphs**, which have no edge connections to the original graph (Appendix E.5.1, Fig. 13, page 23 of the paper).
> - **Extended subgraphs**, which remain connected to the original graph (Appendix E.5.2, Fig. 14, page 24 of the paper).
>
> **Therefore, only Fig. 14 (note: this figure corresponded to Fig. 13 in the initial version of the paper) includes dashed curves to denote extended subgraphs, emphasizing the distinction between these two scenarios.**
>
> For the experimental setting of anomaly detection performance on extended subgraphs scenario (E.5.2), we simulate the extended-subgraph scenario by selecting the largest connected component of the original graph and partitioning it into two subgraphs, $\mathcal{G} _ 1$ and $\mathcal{G} _ 2$. We use a 9:1 node ratio to minimize structural disruption. We then compare two training schemes:
> - **Full-graph training**: training the entire model on the complete graph and testing on $\mathcal{G} _ 2$.
> - **Partial-graph transfer**: train only on $\mathcal{G} _ 1$ and directly transfer the trained MLP to $\mathcal{G} _ 2$ for testing.
>
> This setup evaluates whether the MLP can generalize its structure-aware representations to graph regions that were unseen during training.
>
> Regarding the comparison with HUGE: this method jointly trains the MLP and GNN using globally heterogeneous-guided ranking and alignment losses. **However, when transferring to the new subgraph, HUGE lacks a heterogeneity-aware metric for the new subgraph, leading to degrade structural perception and reduced anomaly detection performance.** As shown in Fig. 13 (page 24 of the paper), our method achieves superior transferability. This conclusion is further supported by the strong performance that HUGE attains on the ACM dataset under full-graph training, indicating that its performance collapses mainly during transfer.
>
> Thank you again for your valuable comments and suggestions. If you have any further questions, we would be pleased to discuss them with you.

---

### Official Review · Reviewer_r86T · 2025-11-01

**Soundness:** 2
**Presentation:** 2
**Contribution:** 2
**Rating:** 4
**Confidence:** 4

**Summary:**

The paper introduces SCALER, a self-supervised graph anomaly detection framework that combines MLP and GNN. SCALER achieves high performance and fast inference without costly GNN aggregation, even a large-scale OGB dataset. It uses a dual-level contrastive learning network, leveraging both node-level and edge-level contrast to train the MLP. Additionally, SCALER includes a neighborhood entropy-guided anomaly score correction module to enhance robustness.

**Strengths:**

* The framework is well-designed and comprehensively evaluated. The experiments cover multiple datasets and metrics, demonstrating the effectiveness and efficiency of SCALER.
* The neighborhood entropy-guided anomaly score correction module is interesting to me.
* The proposed method avoids costly GNN neighbor aggregation during inference, thus speeding up the process and reducing GPU memory usage, especially for large-scale datasets. This is crucial for practical deployment.

**Weaknesses:**

* In Fig. 3, is the "structure-aware MLP" the same as the "MLP" used in training? Both are shown in orange boxes. It would be helpful to add clear annotations to the figure. Also, why use different names? This would make it easier for readers to understand.
* The paper doesn't seem to mention any work on accelerating GNN aggregation. This relevant work should be discussed.
* Have there been attempts at better methods for the global negative edge pool, such as adaptive numbers and targeted selection instead of fixed numbers and random selection?
* In Table 1, why are the AUPRC for the comparison methods on the Reddit and YelpChi datasets very low? This doesn't match the results in [1].

[1] GCTAM: Global and Contextual Truncated Affinity Combined Maximization Model For Unsupervised Graph Anomaly Detection. International Joint Conference on Artificial Intelligence (2025).

**Questions:**

See Weaknesses.

---

> ### Author Response · Authors · 2025-11-20
> **Response to Reviewer r86T ---- Part 1/2**
>
> Thanks a lot for your insightful comments and detailed suggestions, which have been very helpful in improving our paper. We hope our responses address all the points you raised.
> > **Q1.** Clarifying the Difference Between "MLP" and "Structure-Aware MLP" in Fig. 3.
>
> We appreciate your suggestion. **"MLP" and "Structure-Aware MLP" actually refer to the same model.** We used different names initially in an attempt to distinguish the MLP before training from the MLP after training.
>
> We agree that this distinction was not sufficiently clear in the current figure. In the revised version (Fig. 3 on page 4 of the paper), we will add clearer annotations and unify the terminology to avoid confusion and improve readability.
>
> > **Q2.** The paper doesn't seem to mention any work on accelerating GNN aggregation. This relevant work should be discussed.
>
> Thanks for this valuable suggestion. In the original version, our focus was primarily on unsupervised GAD frameworks, and we did not discuss techniques for accelerating GNN aggregation. **We agree that this is an important and highly relevant line of research. However, since our work mainly focuses on accelerating GNN inference, we have added a discussion of GNN aggregation acceleration techniques in Appendix B RELATED WORK (page 12-13 of the paper ).**
>
> GNN inference acceleration techniques typically include quantization, pruning, and knowledge distillation (KD) [1]. Quantization compresses continuous values into compact numerical representations, while pruning removes redundant parameters to reduce computational overhead. KD transfers knowledge from a large teacher model to a smaller student model. Since message passing is the primary computational bottleneck during inference, several recent studies explore transferring the knowledge of trained GNNs into MLPs [1]. For example, GLNN [2] adopts a standard KD pipeline in which an MLP student mimics a GNN teacher, enabling graph-free inference but inevitably weakening the student's ability to capture structural information. Graph-MLP [3] incorporates structural cues through a neighbor-contrastive loss. NOSMOG [4] strengthens structural awareness by injecting positional features and enforcing representational similarity with GNN teachers. GraphECL [5] guides the MLP to learn neighborhood structural distributions distilled from a GNN. SimMLP [6] introduces a self-supervised framework that aligns GNN and MLP embeddings to inject structural information into MLPs, enabling GNN-level performance with dramatically faster inference.
>
> However, most existing methods in GAD follow CoLA's node-subgraph contrastive framework, which requires GNN aggregation and multiple rounds of subgraph sampling during inference. This leads to slow inference and limits their applicability in real-world settings. Motivated by KD, we depart from the CoLA framework and adopt a distillation-based paradigm to GAD. Through our MLP–GNN joint training framework, we obtain a structure-aware MLP for anomaly evaluation, completely eliminating the time costs associated with GNN aggregation and multiple sampling.
>
> **References**
>
> [1] Lu Ma, et al. Acceleration algorithms in gnns: A survey. In TKDE 2025.
>
> [2] Shichang Zhang, et al. Graph-less neural networks: Teaching old mlps new tricks via distillation. In ICLR 2022.
>
> [3] Yang Hu, et al. Graph-MLP: node classification without message passing in graph. In arXiv 2021.
>
> [4] Yijun Tian, et al. Learning MLPs on graphs: A unified view of effectiveness, robustness, and efficiency. In ICLR 2023.
>
> [5] Teng Xiao, et al. Efficient contrastive learning for fast and accurate inference on graphs. In ICML 2024.
>
> [6] Zehong Wang, et al. Training mlps on graphs without supervision. In WSDM 2025.

---

> ### Author Response · Authors · 2025-11-20
> **Response to Reviewer r86T ---- Part 2/2**
>
> > **Q3.** Have there been attempts at better methods for the global negative edge pool, such as adaptive numbers and targeted selection instead of fixed numbers and random selection?
>
> Thanks for this insightful question.
>
> Our method constructs a global negative edge pool by randomly sampling a fixed number of edges. This design avoids per-positive negative mining and ensures that the framework remains highly efficient and scalable.
>
> To address your question, we further explored several alternative strategies, including:
>
> - **Adaptive sampling size based on batch edge count.** For each batch, we sample a fixed proportion \(c\) of the count of edges in that batch as negatives: $c \cdot |\mathcal{E}_{batch}|, \quad c=0.05 $ in our experiment.
> - **Adaptive sampling size based on batch mean degree.** For each batch, the number of negative samples is determined by the mean degree $\bar{d}$ in that batch: $c \cdot \bar{d}, \quad c=10 $ in our experiment.
> - **Top-K targeted sampling based on edge probability.** For each batch, we compute edge probabilities based on node-to-node similarity and select the Top-K highest-probability edges as negatives. In our experiments, K is set to 60.
> - **Similarity-biased probability sampling.** Negative edges are sampled with probabilities proportional to node-to-node similarity. The number of negatives is fixed to 60.
>
> The results are shown in the table below. While some of these variants offer marginal performance improvements, they introduced additional computational overhead due to the more complexity in sampling. **Overall, the fixed-size random sampling strategy offers the best balance of effectiveness, efficiency, and scalability, and therefore remains our preferred choice.**
>
> | Datasets  | Edge count  | Mean degree  | Top-K | Probability sampling | **Proposed** |
> | --------     | ------ | ------  | ------  | ------  | ------  |
> | Citeseer    | 0.9603 | 0.9635  | **0.9639** | 0.9628 | 0.9621  |
> | Citation    | 0.9535 | 0.9566  | **0.9568** | 0.9550 | 0.9557  |
>
> > **Q4.** In Table 1, why are the AUPRC for the comparison methods on the Reddit and YelpChi datasets very low? This doesn't match the results in [1].
>
> We found that the average AUPRC values for the Reddit and YelpChi baselines differed by only 0.003 and 0.005, respectively, compared to the reported results. Such minor discrepancies may arise from variations in implementation environment, datasets processing, or random seed configurations. Importantly, even under conditions where this comparison method yields slightly higher values, our approach still outperforms those mthods.
>
> Thank you again for your valuable comments and suggestions. If you have any further questions, we would be pleased to discuss them with you.

---

### Official Review · Reviewer_CCsg · 2025-11-03

**Soundness:** 3
**Presentation:** 3
**Contribution:** 3
**Rating:** 6
**Confidence:** 3

**Summary:**

This paper proposes SCALER, a UGAD framework that transfers structural knowledge from GNNs into MLPs, combined with dual-level contrastive learning (node- and edge-level) and entropy-guided anomaly score correction. The goal is to retain strong anomaly detection performance while achieving fast inference without GNN aggregation. Experiments on benchmark datasets demonstrate improved AUROC/AUPRC and significantly faster inference compared to existing methods.

**Strengths:**

S1. This paper addresses a practical UGAD bottleneck by enabling fast inference without GNN aggregation.

S2. It proposes a dual-level contrastive learning design that effectively improves over existing subgraph contrast approaches (e.g., CoLA).

S3. It provides strong empirical results, showing consistent performance gains and large inference speedups across various real-world datasets.

**Weaknesses:**

W1. This paper shows limited novelty, as it mainly refines existing ideas rather than introducing a fundamentally new UGAD approach.

W2. This paper uses relatively high anomaly ratios (3-6%), which weakens claims of real-world applicability.

W3. This paper lacks training-time comparisons, despite requiring a GNN and dual contrastive objectives during training.

W4. This paper relies on a heuristic entropy-based score correction without validating alternatives.

**Questions:**

Q1. What is the core novelty that prior node-neighborhood contrast and GNN-to-MLP transfer methods do not already provide?

Q2. Can you report results under much lower anomaly ratios (e.g., <1%) to support real-world applicability?

Q3. How does the training time and memory compare to baselines, given the need for a GNN and dual contrast during training?

Q4. Why use a heuristic entropy-based correction, and how does it compare to a simple learnable alternative or show robustness via sensitivity analysis?

---

> ### Author Response · Authors · 2025-11-20
> **Response to Reviewer CCsg ---- Part 1/2**
>
> We greatly appreciate your valuable time and constructive comments, and we hope our responses fully address your concerns.
> > **Q1.** What is the core novelty that prior node-neighborhood contrast and GNN-to-MLP transfer methods do not already provide?
>
> Our core contributions can be summarized into two key innovations:
> - **A Distillation-Inspired Innovation to the Traditional UGAD Framework for Fast, Aggregation-Free and Subgraph-Sampling-Free Inference.** Current unsupervised graph anomaly detection (UGAD) methods can be roughly divided into contrastive-based and reconstruction-based approaches. Our work focuses on the former. The pioneering method, CoLA, introduced contrastive learning to UGAD by constructing node–subgraph positive and negative pairs to capture the relationships between nodes and their local structures. Many subsequent works (e.g., ANEMONE, SL-GAD, Sub-CR, GRADATE, NLGAD, AD-GCL, SAMCL) refine this framework with more sophisticated contrastive designs or detection strategies. However, they suffer from long inference times due to GNN neighborhood aggregation and subgraph sampling. To overcome this limitation, we depart from CoLA's node–subgraph contrast paradigm and propose a distillation-inspired MLP–GNN joint learning framework that efficiently produces a structure-aware MLP for anomaly evaluation without costly aggregation or subgraph sampling. **Our framework is specifically designed for the unsupervised setting, in contrast to most GNN-to-MLP transfer methods that require labeled supervision.**
>
> - **Introducing Edge-Level Contrast via a Dual-Level Synergistic Contrastive Network to Fill the Edge-Level Contrast Gap in GAD Tasks.** Existing graph contrastive learning(GCL) strategies in GAD mainly focus on node-level and subgraph-level contrast, failing to explore edge-level contrast. This gap result in suboptimal anomaly detection performance. (More details on contrastive–based UGAD methods can be found in Appendix B RELATED WORK, page 12-13 of the paper.)
>
> > **Q2.** Can you report results under much lower anomaly ratios (e.g., <1%) to support real-world applicability?
>
> Thank you for raising this important point. We agree that anomaly ratios in real-world settings can be much lower than 3–6%. **Our original experimental setup followed the widely adopted anomaly injection protocol from prior UGAD works**, including CoLA, GADAM, ANEMONE, SL-GAD, and others, which uses 3–6% anomaly ratios as a standard benchmark for fair comparison.
>
> To further address your concern about practical applicability, we conducted additional experiments with much lower anomaly ratios (e.g., <1%). Specifically, we re-injected anomalies into the Citeseer and Citation datasets at 0.9% and 1.0% anomaly ratios, respectively, and compared our method against strong baselines including HUGE, GADAM, and SAMCL. The results in the table below show that even with significantly lower anomaly ratios, our method maintains competitive AUROC. As expected, AUPRC decreases compared to the higher anomaly ratio setting, since weaker anomaly signals make the task more challenging, but our method still achieves a higher AUPRC than all baselines. **These results demonstrate that our approach remains robust and effective in more challenging, real-world anomaly detection scenarios.**
>
> | Datasets | Metrics | SAMCL  | GADAM  | HUGE  |**SCALER (Proposed)** |
> | -------- | ----    | ------ | ------  | ------  | ------  |
> | Citeseer | AUROC   | 0.9310 | 0.9505  | 0.9695  | **0.9818**  |
> |          | AUPRC   | 0.2114 | 0.6151  | 0.6209  | **0.7278**  |
> | Citation | AUROC   | 0.9086 | 0.8338  | 0.8864  | **0.9545**  |
> |          | AUPRC   | 0.1821 | 0.2927  | 0.1917  | **0.4073**  |
>
> **Notes**
>
> All referenced baseline methods, including CoLA, ANEMONE, SL-GAD, Sub-CR, GRADATE, NLGAD, AD-GCL, SAMCL, GADAM, and HUGE, are listed in the Experimental Settings on page 6 of the paper (Section 4.1).

---

> ### Author Response · Authors · 2025-11-20
> **Response to Reviewer CCsg ---- Part 2/2**
>
> > **Q3.** How does the training time and memory compare to baselines, given the need for a GNN and dual contrast during training?
>
> Thank you for drawing our attention to this important aspect.
>
> We analyzed the training time of all methods on Citation, Reddit, and YelpChi dataset. **As shown in the Fig. 6 (page 20 of the paper), our approach remains highly competitive in training time, outperforming most deep learning methods.** Thus, our method strikes a favorable balance among training cost, inference speed, and detection performance.
> Regarding memory usage, as shown by the results in Table 1 (page 7 of the paper), our method successfully handles large-scale OGB datasets, whereas many baseline methods encounter out-of-memory issues. **This demonstrates the strong scalability and practical applicability of our approach.**
>
> Overall, our approach ranks among the top-performing methods in terms of both effectiveness and efficiency.
>
> > **Q4.** Why use a heuristic entropy-based correction, and how does it compare to a simple learnable alternative or show robustness via sensitivity analysis?
>
> The anomaly score for each node is computed based on the consistency between the node and its entire first-order neighborhood. **Our entropy-based anomaly score correction module is applied only during the inference phase: it measures the uncertainty of a node's local structures to provides additional signals for anomaly estimation, making the score more robust.** Similar to NLGAD, which uses the standard deviation of 256 node–subgraph contrastive scores as an uncertainty signal, our method utilizes the neighbor entropy to capture the reliability of the node's local structural context. Because real neighborhoods may include anomalous or noisy neighbors, the entropy signal effectively indicates the neighborhood uncertainty and thus provides a meaningful correction to the anomaly score.
>
> Regarding sensitivity analysis, as shown in Fig. 4 (page 8 of the paper), introducing the unified correction module consistently improves performance across all datasets compared to the uncorrected version, demonstrating the robustness of this module.
>
> Furthermore, following your suggestion, we implemented a simple learnable alternative by replacing the neighbor-entropy signal with a one-layer MLP when computing the node–neighbor consistency score during training. The results in the table below show that this learnable alternative yields lower performance and shows larger variance across runs compared to our method, **confirming that the entropy-based correction offers a more stable and reliable solution in the unsupervised setting.**
>
> | Datasets | Metrics | SCALER w/ learnable |**SCALER w/ neighborhood entropy (Proposed)** |
> | -------- | ----    | ------ | ------  |
> | DBLP     | AUROC   | **0.9356 ± 0.44** | 0.9326 ± 0.27  |
> |          | AUPRC   | 0.6424 ± 1.34 | **0.6646 ± 1.17**  |
> | YelpChi  | AUROC   | 0.6026 ± 0.85 | **0.6211 ± 0.26**  |
> |          | AUPRC   | 0.0735 ± 0.27 | **0.0806 ± 0.15**  |
>
> Thank you again for your valuable comments and suggestions. If you have any further questions, we would be pleased to discuss them with you.

---

### Author Response · Authors · 2025-12-03
**Official Comment by Authors**

Dear Area Chair and Reviewers,

We sincerely appreciate your time, efforts, and constructive feedback on our submission. Given the number and richness of the reviews, we summarize below the main points raised by the reviewers and our corresponding responses.

> **Strengths**

- The paper addresses a practical bottleneck in UGAD by decoupling costly GNN aggregation from inference, thereby enabling efficient MLP-only inference ($\color{Magenta}{All}$ Reviewers).

- It proposes a novel dual-level contrastive learning network that fills the edge-level contrast gap in GAD task (Reviewer $\color{red}{CCsg}$, $\color{green}{ndWk}$).

- It introduces an interesting neighborhood entropy-guided anomaly score correction module (Reviewer $\color{orange}{r86T}$, $\color{Blue}{pyNh}$).

- The overall framework is well-designed (Reviewer $\color{orange}{r86T}$), with comprehensive experimental design and a valuable generalization analysis in Appendix E.5 (Reviewer $\color{green}{ndWk}$, $\color{Blue}{pyNh}$).

- The proposed method achieves strong empirical results, delivering high detection performance and substantial inference speedups, while scaling effectively to large datasets—an important factor for practical deployment ($\color{Magenta}{All}$ Reviewers).

> **Responses**

- **Clarification** **of** **differences** **from** **prior** **methods:** We first provide a detailed review of existing UGAD frameworks and GCL strategies, and then highlight two key innovations that clearly distinguish our approach from prior methods (**Q1**, Reviewer $\color{red}{CCsg}$, and **W1**, Reviewer $\color{Blue}{pyNh}$).

- **Five** **additional** **experiments:** We evaluate much lower anomaly ratios (<1%) performance (**Q2**, Reviewer $\color{red}{CCsg}$), report training time/memory comparisons (**Q3**, Reviewer $\color{red}{CCsg}$), and validate the entropy-guided correction via sensitivity analysis, and a learnable alternative (**Q4**, Reviewer $\color{red}{CCsg}$). We further explore four improved designs for the global negative edge pool (**W3**, Reviewer $\color{orange}{r86T}$) and four complex edge representation methods (**Q3**, Reviewer $\color{green}{ndWk}$).

- **Enhancements** **to** **the** **paper:** We have revised Fig. 3 to eliminate the ambiguity in the previous version (**W1**, Reviewer $\color{orange}{r86T}$) and improved the description of generalization analysis in the new version (**W2**, Reviewer $\color{green}{ndWk}$). And We have added a discussion of inference acceleration on GNNs in Appendix B RELATED WORK (**W2**, Reviewer $\color{orange}{r86T}$).

- **Further** **discussion** **of** **our** **method:** We clarify the meaning of "dual-level synergistic" (**W2**, Reviewer $\color{Blue}{pyNh}$) and explore the potential of subgraph-level contrastive learning in GAD (**Q1**, Reviewer $\color{Blue}{pyNh}$). And we further explain, with support from related literature, the applicability of our method on graphs with long-range dependencies (**W3**, Reviewer $\color{Blue}{pyNh}$).

- **Further** **discussion** **of** **experimental** **results:** We clarify the small gap between our AUPRC results and the reviewer-cited reference, and provide a reasonable error analysis (**W4**, Reviewer $\color{orange}{r86T}$). We further clarify the AUROC–AUPRC discrepancy observed on Reddit is reasonable (**Q1**, Reviewer $\color{green}{ndWk}$), and discuss why $\mathcal{L}_{\text{reg}}$ is not suitable for organic anomalies from the perspective of anomaly sources (**Q2**, Reviewer $\color{green}{ndWk}$).

We have endeavored to respond proactively and comprehensively to all reviewers’ questions and concerns. Once again, we deeply appreciate the time and expertise you have invested in our work. Your encouraging feedback motivates us to further improve and advance this line of research for the broader community.

Best regards,

Authors of Paper 1223

---

### Meta-Review · Area_Chair_Ummx · 2025-12-22

**Summary:**

This work introduces SCALER, a self-supervised framework for unsupervised GAD, aiming to reduce the inference-time overhead of GNN-based methods by avoiding neighborhood aggregation at deployment. The approach relies on a dual-level contrastive learning strategy, combining node-level and edge-level signals to transfer structural information from a GNN to a structure-aware MLP, additionally supported by a heuristic entropy-based score correction to improve the robustness in GAD. Empirical results on multiple GAD datasets suggest that SCALER can improve the detection performance of 13 state-of-the-art baselines.

**Reviewer Concerns:**

Many weaknesses and concerns are raised. The AC summarizes the major ones below, alongside with how the authors address them.
- **Novelty and Differentiation from Prior Work.** Reviewers question the paper’s core novelty, arguing that the proposed dual-level contrastive learning is largely built upon existing graph contrastive learning (GCL)  ideas in GAD, such as those in CoLA and multi-scale GCL methods, rather than introducing a fundamentally new approach. The rebuttal clarifies the differences in the motivation and the design (mainly in the edge-level contrast).
- **Experimental Setup and Real-World Applicability.** Concerns are raised about realistic evaluation settings, particularly the use of relatively high anomaly ratios (3-6%), which may not reflect real-world scenarios. The rebuttal clarifies that the setting follows the cmmonly used evaluation protocol in prior GAD studies. For the real-world applicability, the rebuttal also highlights the use of the protocol that includes synthetic anomaly injection into node classification datasets for evaluating GAD performance. However, this discards many real-world GAD datasets that have real anomalies. It is unclear why we should prefer these synthetic datasets, given the availability of the real datasets. It is argued in the GAD community that many GCL-based methods implicitly leverage the mechanisms/protocols in the anomaly injection strategies, which may have led to data leakage issues.
- **Inconsistent Reported Results.** Reviewers question the inconsistencies of the comparison methods on the Reddit and YelpChi datasets in prior studies and in this work. The rebuttal notes that the inconsistencies are very small, i.e., differ by only 0.003 and 0.005 in AUPRC. The AC cross-checks the results and confirms the small differences. However, the work seems to use a very different set of comparison methods from the mentioned IJCAI'25 work. The experiments seem to focus on comparing with GCL-based methods. Reviewers also raise the necessity of comparing to GNN acceleration methods. The rebuttal discusses the differences, but it does not link the acceleration and GAD methods together and discuss the advantages of the proposed approach w.r.t. those combined methods.
- **Design Choices.** Several components are considered as under-justified, such as the entropy-based score correction, global negative edge sampling, and simple edge embedding definitions. The rebuttal provides additional results on (DBLP, YelpChi) or (CiteSeer and Citation) to compare the designs in the paper with alternative designs.

The AC concludes that (1) the claim on the novelty of the work is not convincing enough, as it does not provide much additional insight into the GAD problem, and (2) the empirical justification is weak due to the focus on the anomaly injection-based evaluation protocol. The other concerns have been mostly addressed, though the design choices and its advantages over GNN acceleration alternatives should be more comprehensively evaluated.

**Reviewer Scores:**

The paper gains four reviews, including two weak accept ratings and two weak reject ratings. The two reject ratings are not likely to change due to their concerns over the novelty, model designs, and their justification, since these concerns are not satisfactorily addressed in the rebuttal. Besides, one weak accept review also questions the novelty of the work.

---

### Decision · Program_Chairs · 2026-01-26

Reject